# Tree-Sliced Wasserstein Distance with Nonlinear Projection

**Thanh Tran** [* 1]  **Viet-Hoang Tran** [* 2]  **Thanh Chu** [2]  **Trang Pham** [3]  **Laurent El Ghaoui** [† 1]  **Tam Le** [† 4]
**Tan M. Nguyen** [† 2]

## Abstract

Tree-Sliced methods have recently emerged as an alternative to the traditional Sliced Wasserstein (SW) distance, replacing one-dimensional lines with tree-based metric spaces and incorporating a splitting mechanism for projecting measures. This approach enhances the ability to capture the topological structures of integration domains in Sliced Optimal Transport while maintaining low computational costs. Building on this foundation, we propose a novel nonlinear projectional framework for the Tree-Sliced Wasserstein (TSW) distance, substituting the linear projections in earlier versions with general projections, while ensuring the injectivity of the associated Radon Transform and preserving the well-definedness of the resulting metric. By designing appropriate projections, we construct efficient metrics for measures on both Euclidean spaces and spheres. Finally, we validate our proposed metric through extensive numerical experiments for Euclidean and spherical datasets. Applications include gradient flows, self-supervised learning, and generative models, where our methods demonstrate significant improvements over recent SW and TSW variants. The code is publicly available at https://github.com/thanhqt2002/NonlinearTSW.

## 1. Introduction

Optimal Transport (OT) (Villani, 2008; Peyré et al., 2019) and Sliced-Wasserstein (SW) (Rabin et al., 2011; Bonneel et al., 2015) provide geometrically meaningful metrics in the space of probability measures, making them widely applicable across various fields. These include machine learning (Nguyen et al., 2021b; Bunne et al., 2022; Hua et al., 2023; Fan et al., 2022; Le et al., 2024a;b; Kessler et al., 2025; Chapel et al., 2025; Chapel & Tavenard, 2025), multimodal data analysis (Park et al., 2024; Luong et al., 2024), computer vision and graphics (Lavenant et al., 2018; Nguyen et al., 2021a; Saleh et al., 2022; Rabin et al., 2011; Solomon et al., 2015; Vu et al., 2025), statistics (Mena & Niles-Weed, 2019; Weed & Berthet, 2019; Wang et al., 2022; Pham et al., 2024; Liu et al., 2022; Nguyen et al., 2022; Nietert et al., 2022). By utilizing the closed-form solution of one-dimensional OT problems, SW substantially reduces the computational complexity typically associated with OT (Rabin et al., 2011; Bonneel et al., 2015; Peyré et al., 2019).

**Related work.** Several enhancements have been proposed to improve different aspects of the SW framework, including optimizing the sampling process (Nguyen et al., 2020; Nadjahi et al., 2021; Nguyen et al., 2024a), selecting optimal projection directions (Deshpande et al., 2019), and refining the projection mechanism (Kolouri et al., 2019; Chen et al., 2022; Bonet et al., 2023).

The Tree-Sliced framework (Tran et al., 2025c;a;b) has recently emerged as an alternative to the traditional SW framework by replacing one-dimensional projection lines with more structured domains, known as tree systems. These systems function similarly to lines but incorporate a more sophisticated and interconnected structure. Instead of projecting measures onto individual lines, this method distributes them across multiple linked lines, forming a hierarchical structure. This approach enhances the representation of topological information while preserving the computational efficiency by leveraging the closed-form solution of OT problems in tree-metric spaces (Indyk & Thaper, 2003; Le et al., 2019). However, Tree-Sliced frameworks are still restricted to linear projections, as the integration domains used in (Tran et al., 2025c;b) remain confined to hyperplanes. In contrast, several advanced projection techniques have been investigated for SW, enhancing its flexibility and effectiveness (Kuchment, 2006; Kolouri et al., 2019; Chen et al., 2022; Bonet et al., 2023). Building on these advancements, this paper introduces a novel framework for the tree-sliced method that integrates nonlinear projections, further broad-

---

[*]Equal contribution [†]Co-last authors [1]VinUniversity [2]National University of Singapore [3]Movian AI [4]The Institute of Statistical Mathematics. Correspondence to: Viet-Hoang Tran <hoang.tranviet@u.nus.edu>.

*Proceedings of the 42$^{nd}$ International Conference on Machine Learning*, Vancouver, Canada. PMLR 267, 2025. Copyright 2025 by the author(s).

ening its scope and improving its performance.

**Contribution.** Our contributions are three-fold:

- We introduce the Generalized Radon Transform and Spatial Radon Transform on Systems of Lines, extending previous variants that were limited to linear projections. Along with these extensions, we provide theoretical results and proofs demonstrating their injectivity. Furthermore, we generalize the conventional Euclidean framework by proposing a spherical tree-sliced version tailored for nonlinear Radon transforms.

- We introduce Tree-Sliced distances based on the proposed Radon Transform, namely the Circular Tree-Sliced Wasserstein distance and the Spatial Tree-Sliced Wasserstein distance, along with an extended version for the spherical setting, called the Spatial Spherical Tree-Sliced Wasserstein distance. Furthermore, we examine different choices of functions that define the nonlinear projections, analyze their computational complexity and explain why certain choices lead to more efficient metrics.

- We assess the proposed metrics across various tasks, including gradient flows and generative models, on both Euclidean and spherical data, highlighting their practical effectiveness and computational efficiency.

**Organization.** The structure of the paper is as follows: Section 2 provides an overview of different variants of the Wasserstein distance, while Section 3 explores various forms of the Radon Transform with Nonlinear Projection used to derive corresponding Wasserstein distances. Section 4 introduces novel Radon Transforms on Systems of Lines with Nonlinear Projection and examines their injectivity. In Section 5, two new Tree-Sliced Wasserstein distances associated with the proposed transform are introduced, along with an analysis of their fundamental components. Finally, Section 6 assesses the performance of the proposed methods on both Euclidean and Spherical data. Additional materials related to the spherical settings of the proposed method, as well as background, theoretical foundations, and supplementary content, are provided in the Appendix.

## 2. Preliminaries

This section reviews the Wasserstein distance between measures and its various sliced variants. For simplicity, the focus is on measures with a finite first moment, while measures with a finite $p^{\text{th}}$-moment are treated analogously.

**Wasserstein distance.** Given a measurable space $\Omega$ endowed with a metric $d$. Let $\mu, \nu \in \mathcal{P}(\Omega)$, and $\mathcal{P}(\mu, \nu)$ be the set of distributions $\pi$ coupling between $\mu$ and $\nu$. The

Wasserstein distance (W) (Villani, 2008) between $\mu, \nu$ is:

$$W(\mu, \nu) = \inf_{\pi \in \mathcal{P}(\mu, \nu)} \int_{\Omega \times \Omega} d(x, y) \, d\pi(x, y). \quad (1)$$

**Sliced Wasserstein distance.** The Radon Transform (Helgason, 2011) $\mathcal{R} : L^1(\mathbb{R}^d) \to L^1(\mathbb{R} \times \mathbb{S}^{d-1})$ is:

$$\mathcal{R}f(t, \theta) = \int_{\mathbb{R}^d} f(y) \cdot \delta(t - \langle y, \theta \rangle) \, dy, \quad (2)$$

where $\delta$ is the Dirac delta function.

The Sliced Wasserstein distance (SW) (Rabin et al., 2011; Bonneel et al., 2015) between $\mu, \nu \in \mathcal{P}(\mathbb{R}^d)$ is:

$$SW(\mu, \nu) = \int_{\mathbb{S}^{d-1}} W(\mathcal{R}f_\mu(\cdot, \theta), \mathcal{R}f_\nu(\cdot, \theta)) \, d\sigma(\theta), \quad (3)$$

where $\sigma = \mathcal{U}(\mathbb{S}^{d-1})$ is the uniform distribution on the sphere, and $f_\mu, f_\nu$ are the probability density functions of $\mu, \nu$, respectively. The one-dimensional Wasserstein distance in Eq. (3) has the closed-form $W(\theta \sharp \mu, \theta \sharp \nu) = \int_0^1 |F_{\mathcal{R}f_\mu(\cdot, \theta)}^{-1}(z) - F_{\mathcal{R}f_\nu(\cdot, \theta)}^{-1}(z)| dz$, where $F_{\mathcal{R}f_\mu(\cdot, \theta)}, F_{\mathcal{R}f_\nu(\cdot, \theta)}$ are the cumulative distribution functions of $\mathcal{R}f_\mu(\cdot, \theta), \mathcal{R}f_\nu(\cdot, \theta)$, respectively. The Monte Carlo method is employed to approximate the intractable integral in Eq. (3):

$$\widehat{SW}(\mu, \nu) = \frac{1}{L} \sum_{i=1}^{L} W(\mathcal{R}f_\mu(\cdot, \theta_i), \mathcal{R}f_\nu(\cdot, \theta_i)), \quad (4)$$

where $\theta_1, \ldots, \theta_L$ are drawn independently from $\sigma$.

**Tree-Sliced Wasserstein distance on Systems of Lines.** Rather than projecting functions onto lines, i.e., directions in $\mathbb{S}^{d-1}$, as in the original Radon Transform, (Tran et al., 2025c;b) introduces an alternative approach that replaces lines with different metric measure spaces, known as tree systems. These tree systems are formed by connecting multiple copies of real lines, creating a structured space. Functions are then partitioned using a splitting mechanism and projected onto these spaces, effectively capturing the positional information of both measure supports and slices. Further details on this Tree-Sliced framework can be found in Appendix A and Appendix C.1.

## 3. Radon Transform with Nonlinear Projection

In this section, we explore nonlinear projections utilized in various existing Sliced Wasserstein variants, and compare them with the traditional linear projection in the original Radon Transform that leads to the Sliced Wasserstein distance. Based on these observations, we provide an overview of how these nonlinear projections can be incorporated into the framework of the Radon Transform on Systems of Lines.

## 3.1. Generalized and Spatial Radon Transforms

Let $\Psi$ be a set of feasible parameter, the Generalized Radon Transform $\mathcal{G}$ (GRT) (Kolouri et al., 2019) is defined by:

$$\mathcal{G} : L^1(\mathbb{R}^d) \longrightarrow L^1(\mathbb{R} \times \Psi),$$

$$\text{s.t. } \mathcal{G}f(t, \psi) = \int_{\mathbb{R}^d} f(y) \cdot \delta(t - g(y, \psi)) \, dy. \quad (5)$$

Here, $g \colon \mathbb{R}^d \times \Omega \to \mathbb{R}$ is called the defining function of $\mathcal{G}$. The GRT of $f \in L^1(\mathbb{R}^d)$ is the integration of $f$ over hypersurfaces $\{y \in \mathbb{R}^d : t = g(y, \psi)\}$ for $t \in \mathbb{R}, \psi \in \Psi$. One common choice for the defining function occurs when $\Psi = \mathbb{R}_{\geqslant 0} \times \mathbb{S}^{d-1}$, and $g$ is defined as the circular function:

$$g(y, r, \theta) = \|y - r\theta\|_2, \ \forall y \in \mathbb{R}^d, (r, \theta) \in \Psi. \quad (6)$$

This choice results in the Circular Radon Transform (CRT) (Kuchment, 2006). Another possible defining function, based on homogeneous polynomials of odd degree, is presented in (Rouvière, 2015). However, this represents a special case of the next Radon Transform we discuss. Given a positive integer $d_\theta$, the Spatial Radon Transform $\mathcal{H}$ (SRT) (Chen et al., 2022) is defined by:

$$\mathcal{H} : L^1(\mathbb{R}^d) \longrightarrow L^1(\mathbb{R} \times \mathbb{S}^{d_\theta - 1}),$$

$$\text{s.t. } \mathcal{H}f(t, \theta) = \int_{\mathbb{R}^d} f(y) \cdot \delta(t - \langle h(y), \theta \rangle) \, dy, \quad (7)$$

where $h \colon \mathbb{R}^d \to \mathbb{R}^{d_\theta}$ is an injective continuous map. The SRT of $f \in L^1(\mathbb{R}^d)$ is defined as the integration of $f$ over the hypersurfaces given by $\{y \in \mathbb{R}^d : t = \langle h(y), \theta \rangle\}$ for $t \in \mathbb{R}, \theta \in \mathbb{S}^{d_\theta - 1}$.

*Remark* 3.1. When $\Omega_\theta = \mathbb{S}^{d-1}$, $g(y, \theta) = \langle y, \theta \rangle$ in Eq. (5), or $d = d_\theta, h(y) = y$ in Eq. (7), it is clear that the GRT and SRT recover the original Radon Transform in Eq. (2).

By definition, the SRT is a special case of the GRT, but these two transforms can be interpreted from two distinct perspectives. For a general function $g$, the GRT represents a projection along hypersurfaces in $\mathbb{R}^d$, defined by the level sets of $g$. In contrast, for a general function $h$, the SRT involves mapping functions from $\mathbb{R}^d$ to a new space $\mathbb{R}^{d_\theta}$, where the Radon Transform is then applied.

**Well-definedness and injectivity.** Certain conditions on $g$ in GRT and $h$ in SRT are necessary to ensure the well-definedness of GRT and SRT. Additionally, since injectivity is typically required for Radon Transform variants, specific assumptions on $g$ and $h$ are made to achieve this property. However, since these properties, along with the inverse problem related to Radon Transform variants, remain long-standing research questions (Beylkin, 1984; Ehrenpreis, 2003; Uhlmann, 2003; Homan & Zhou, 2017) and fall beyond the scope of this paper, we restrict our discussion to mentioned examples of $g$ and $h$ found in the literature.

## 3.2. Incorporating Nonlinear Projectional Framework into Systems of Lines Setting

Here, we provide a brief overview of the Radon Transform on Systems of Lines (RTSL) to highlight the potential of integrating the nonlinear approach discussed in Section 3.1. The formal construction of RTSL with notations is detailed in Appendix A and in (Tran et al., 2025c;b). Roughly speaking, unlike the traditional Radon Transform in Eq. (2), which projects onto one line at a time, RTSL extends this concept by simultaneously projecting onto a set of interconnected multiple lines, denoted as $\mathcal{L}$, using a splitting mechanism:

$$\mathcal{R}^\alpha \colon L^1(\mathbb{R}^d) \to \prod_{\mathcal{L} \in \mathbb{L}_k^d} L^1(\mathcal{L}) \quad \text{where} \quad f \mapsto (\mathcal{R}_\mathcal{L}^\alpha f)_{\mathcal{L} \in \mathbb{L}_k^d},$$

$$\text{s.t. } \mathcal{R}_\mathcal{L}^\alpha f(x_i + t \cdot \theta_i)$$

$$= \int_{\mathbb{R}^d} f(y) \cdot \alpha(y, \mathcal{L})_i \cdot \delta(t - \langle y - x_i, \theta_i \rangle) \, dy. \quad (8)$$

Here, $\alpha$ is a continuous map from $\mathbb{R}^d \times \mathbb{L}_k^d$ to $\Delta_{k-1}$ presenting the splitting mechanism, and $(x_i, \theta_i)$ indicates $i^{\text{th}}$-line in $\mathcal{L}$. Eq. (8) can be interpreted as integrating the $i^{\text{th}}$ portion of $f$, given by $f \cdot \alpha_i$, over the hyperplanes $\{y \in \mathbb{R}^d : \langle y, \theta_i \rangle = t + \langle x_i, \theta_i \rangle\}$.

*Remark* 3.2. It is important to note that the partitioning process depends on both $y \in \mathbb{R}^d$ and the set of lines $\mathcal{L}$. This splitting mechanism is absent in the traditional Radon Transform and its variants, as seen in Eqs. (2), (5), (7). This naturally leads to the question:

*Can a nonlinear projectional framework, similar to GRT and SRT, be developed for RTSL?*

Notably, the presence of the map $\alpha$ introduces a trade-off between the effectiveness of the induced Wasserstein metric and the associated theoretical guarantees. Empirical findings indicate that, given the same number of projections (and consequently the same computational cost), the distances derived from RTSL surpass those obtained from the original Radon Transform. However, incorporating $\alpha$ shifts the transformation from operating on straight lines to a more intricate space. This transition may compromise fundamental properties of the transform, such as injectivity, which might no longer be assured in this new framework.

In the next sections, we propose an approach for incorporating the nonlinear projectional framework into the systems of lines setting, thereby generalizing the current RTSL and inducing new variants of the tree-sliced distance.

## 4. Radon Transform on Systems of Lines with Nonlinear Projection

In this section, we extend the current Radon Transform on Systems of Lines (Tran et al., 2025c;b) by incorporating

nonlinear projections. We then analyze key properties of the resulting transforms, including injectivity.

## 4.1. Nonlinear Radon Transform on Systems of Lines

Given a positive integer $k$ representing the number of lines in a tree system, and a continuous splitting map function $\alpha \in \mathcal{C}(\mathbb{R}^d \times \mathbb{L}_k^d, \Delta_{k-1})$ defining the splitting mechanism. Let $\mathcal{L}$ be a system of $k$ lines in $\mathbb{L}_k^d$ and a scalar $r \geqslant 0$. For a function $f \in L^1(\mathbb{R}^d)$, define the function $\mathcal{CR}_{\mathcal{L},r}^\alpha f \in L^1(\mathcal{L})$ as follows:

$$\mathcal{CR}_{\mathcal{L},r}^\alpha f(x_i + t \cdot \theta_i)$$
$$= \int_{\mathbb{R}^d} f(y) \cdot \alpha(y, \mathcal{L})_i \cdot \delta\left(t - \|y - x_i - r\theta_i\|_2\right) \, dy. \quad (9)$$

The *Circular Radon Transform on Systems of Lines* (CRTSL) is defined as the operator:

$$\mathcal{CR}^\alpha : \quad L^1(\mathbb{R}^d) \quad \longrightarrow \quad \prod_{\mathcal{L} \in \mathbb{L}_k^d, r \geqslant 0} L^1(\mathcal{L})$$
$$f \quad \longmapsto \quad \left(\mathcal{CR}_{\mathcal{L},r}^\alpha f\right)_{\mathcal{L} \in \mathbb{L}_k^d, r \geqslant 0}. \quad (10)$$

This is analogous to the CRT described in Section 3.1. In the case of SRT, consider a positive integer $d_\theta$, an injective continuous map $h \colon \mathbb{R}^d \to \mathbb{R}^{d_\theta}$, and a splitting map $\alpha \in \mathcal{C}(\mathbb{R}^{d_\theta} \times \mathbb{L}_k^{d_\theta}, \Delta_{k-1})$. Let $\mathcal{L}$ be a system of lines in $\mathbb{L}_k^{d_\theta}$. For a function $f \in L^1(\mathbb{R}^d)$, define the function $\mathcal{H}_{\mathcal{L}}^\alpha$ as:

$$\mathcal{H}_{\mathcal{L}}^\alpha f(x_i + t \cdot \theta_i) \quad (11)$$
$$= \int_{\mathbb{R}^d} f(y) \cdot \alpha(h(y), \mathcal{L})_i \cdot \delta\left(t - \langle h(y) - x_i, \theta_i \rangle\right) \, dy.$$

The *Spatial Radon Transform on Systems of Lines* (SRTSL) is defined as the operator:

$$\mathcal{H}^\alpha : \quad L^1(\mathbb{R}^d) \quad \longrightarrow \quad \prod_{\mathcal{L} \in \mathbb{L}_k^{d_\theta}} L^1(\mathcal{L})$$
$$f \quad \longmapsto \quad \left(\mathcal{H}_{\mathcal{L}}^\alpha f\right)_{\mathcal{L} \in \mathbb{L}_k^{d_\theta}}. \quad (12)$$

*Remark* 4.1. It is important to note that when the system of lines consists of a single line, i.e. $k = 1$, $\mathcal{CR}^\alpha$ and $\mathcal{H}^\alpha$ recover the GRT and the SRT, respectively.

Properties of $\mathcal{CR}^\alpha$ and $\mathcal{H}^\alpha$ are discussed in the next part.

## 4.2. Well-definedness and Injectivity

**Well-definedness.** Given the setting of $\mathcal{CR}^\alpha$ and $\mathcal{H}^\alpha$ as defined in Eqs. (9), (11), we have $\mathcal{CR}_{\mathcal{L},r}^\alpha f \in L^1(\mathcal{L})$ and $\mathcal{H}_{\mathcal{L}}^\alpha f \in L^1(\mathcal{L})$ for $f \in L^1(\mathbb{R}^d)$. Furthermore, we have the bounds:

$$\|\mathcal{CR}_{\mathcal{L},r}^\alpha f\|_{\mathcal{L}} \leqslant \|f\|_1 \quad \text{and} \quad \|\mathcal{H}_{\mathcal{L}}^\alpha f\|_{\mathcal{L}} \leqslant \|f\|_1. \quad (13)$$

The proofs for these properties are provided in Appendices B.1 and B.2. Additionally, these proofs imply that if $f \in \mathcal{P}(\mathbb{R}^d)$, then $\mathcal{CR}_{\mathcal{L},r}^\alpha f, \mathcal{H}_{\mathcal{L}}^\alpha f \in \mathcal{P}(\mathcal{L})$.

**Injectivity.** For injectivity of $\mathcal{CR}^\alpha$ and $\mathcal{H}^\alpha$, we refer to the concept of $\mathrm{E}(d)$-invariance in splitting maps as introduced in (Tran et al., 2025b). The group $\mathrm{E}(d)$ represents the Euclidean group, which consists of all transformations of Euclidean space $\mathbb{R}^d$ that preserve the Euclidean distance between any two points. Through the canonical action of $\mathrm{E}(d)$ on $\mathbb{R}^d$, an induced action of $\mathrm{E}(d)$ on $\mathbb{L}_k^d$ follows. Appendix A provides a formal description of the underlying group actions associated with these equivariant constructions. A splitting map $\alpha \in \mathcal{C}(\mathbb{R}^d \times \mathbb{L}_k^d, \Delta_{k-1})$ is $\mathrm{E}(d)$-invariant if:

$$\alpha(gy, g\mathcal{L}) = \alpha(y, \mathcal{L}), \quad (14)$$

for all $(y, \mathcal{L}) \in \mathbb{R}^d \times \mathbb{L}_k^d$ and $g \in \mathrm{E}(d)$. We have two results about injectivity of operator $\mathcal{CR}^\alpha$ and $\mathcal{H}^\alpha$.

**Theorem 4.2.** *For an* $\mathrm{E}(d)-$*invariant splitting map* $\alpha \in \mathcal{C}(\mathbb{R}^d \times \mathbb{L}_k^d, \Delta_{k-1})$, $\mathcal{CR}^\alpha$ *is injective.*

**Theorem 4.3.** *For an* $\mathrm{E}(d_\theta)-$*invariant splitting map* $\alpha \in \mathcal{C}(\mathbb{R}^{d_\theta} \times \mathbb{L}_k^{d_\theta}, \Delta_{k-1})$, $\mathcal{H}^\alpha$ *is injective.*

The proofs of Theorem 4.2 and Theorem 4.3 are provided in Appendices B.3 and B.4. Intuitively, the reason why $\mathrm{E}(d)-$invariance in splitting maps ensures the injectivity of $\mathcal{CR}^\alpha$ stems from the fact that the circular defining function in Eq. (6) is primarily based on the Euclidean norm $\|\cdot\|_2$, which itself is an $\mathrm{E}(d)-$ function. Similarly, the reason why $\mathrm{E}(d_\theta)$-invariance in splitting maps guarantees the injectivity of $\mathcal{H}^\alpha$ is that this property is essential for achieving injectivity in the standard Radon Transform on Systems of Lines in $\mathbb{R}^{d_\theta}$, as discussed in (Tran et al., 2025b).

## 4.3. Spatial Spherical Radon Transform on Spherical Trees

In (Tran et al., 2025a), the tree-sliced framework is extended to functions defined on hyperspheres. The techniques presented in Sections 4.1 and 4.2 can be adapted to the spherical setting. We provide a brief derivation here, while a more detailed background and notation are given in Appendix C.1. For $f \in L^1(\mathbb{S}^d)$, we recall the Spherical Radon Transform on Spherical Trees (SRTST), which transforms $f$ to $\mathcal{R}_{\mathcal{T}}^\alpha f \in L^1(\mathcal{T})$, where:

$$\mathcal{R}_{\mathcal{T}}^\alpha f(t, r_{y_i}^x)$$
$$= \int_{\mathbb{S}^d} f(y) \cdot \alpha(y, \mathcal{T})_i \cdot \delta(t - \arccos\langle x, y \rangle) \, dy, \quad (15)$$

Since the literature on defining functions for GRT on the sphere is limited, we focus only on the spatial version of SRTST. Given a positive integer $d_\theta$ and an injective continuous map $h \colon \mathbb{S}^d \to \mathbb{S}^{d_\theta}$, the *Spatial Spherical Radon Transform on Spherical Trees* transforms $f \in L^1(\mathbb{S}^d)$ to

$\mathcal{H}_{\mathcal{T}}^{\alpha} f \in L^1(\mathcal{T})$, where:

$$\mathcal{H}_{\mathcal{T}}^{\alpha} f(t, r_{y_i}^x) \tag{16}$$

$$= \int_{\mathbb{S}^d} f(y) \cdot \alpha(h(y), \mathcal{T})_i \cdot \delta(t - \arccos \langle x, h(y) \rangle) \, dy.$$

As stated in (Tran et al., 2025a), $O(d+1)$-invariance is a necessary property of the splitting map $\alpha$ to ensure the injectivity of $\mathcal{R}^{\alpha}$. A similar result holds in our setting, as described below.

**Theorem 4.4.** *For an $O(d_\theta + 1)$-invariant splitting map $\alpha \in \mathcal{C}(\mathbb{S}^{d_\theta} \times \mathbb{T}_k^{d_\theta}, \Delta_{k-1})$, $\mathcal{H}^{\alpha}$ is injective.*

The detailed derivation of the Spatial Spherical Radon Transform on Spherical Trees and the proof of Theorem 4.4 are provided in Appendices C.2 and C.3.

# 5. Tree-Sliced Wasserstein Distance with Nonlinear Projection

In this section, we propose new distance between measures derived from the variants of the Radon Transform introduced in Section 4. We also examine different choices of functions that define the nonlinear projections and explain why certain choices lead to more efficient metrics.

## 5.1. Definition of Tree-Sliced Distances

For two probability measures $\mu$ and $\nu$ with density function $f_\mu$ and $f_\nu$, and a fixed $r \geq 0$, the *Circular Tree-Sliced Wasserstein Distance* (CircularTSW) between $\mu$ and $\nu$ is defined as the average Wasserstein distance on the tree-metric space $\mathcal{L}$ between the CRTSL of $f_\mu$ and $f_\nu$. Following (Tran et al., 2025c;b), this averaging is taken over the space of trees $\mathbb{T}_k^d \subset \mathbb{L}_k^d$, according to a distribution $\sigma$ on $\mathbb{T}_k^d$ which arises from the tree sampling process.

**Definition 5.1.** The *Circular Tree-Sliced Wasserstein Distance* between $\mu$ and $\nu$ in $\mathcal{P}(\mathbb{R}^d)$ is defined by:

CircularTSW$(\mu, \nu)$

$$:= \int_{\mathbb{T}_k^d} \mathrm{W}(\mathcal{CR}_{\mathcal{L},r}^{\alpha} f_\mu, \mathcal{CR}_{\mathcal{L},r}^{\alpha} f_\nu) \, d\sigma(\mathcal{L}). \tag{17}$$

Similarly, given a choice of the continuous injective map $h \colon \mathbb{R}^d \to \mathbb{R}^{d_\theta}$ in SRTSL, we have the definition of the *Spatial Tree-Sliced Wasserstein Distance* (SpatialTSW) between $\mu$ and $\nu$.

**Definition 5.2.** The *Spatial Tree-Sliced Wasserstein Distance* between $\mu$ and $\nu$ in $\mathcal{P}(\mathbb{R}^d)$ is defined by:

SpatialTSW$(\mu, \nu) := \int_{\mathbb{T}_k^{d_\theta}} \mathrm{W}(\mathcal{H}_{\mathcal{L}}^{\alpha} f_\mu, \mathcal{H}_{\mathcal{L}}^{\alpha} f_\nu) \, d\sigma(\mathcal{L}). \tag{18}$

Both CircularTSW and SpatialTSW distances are, indeed, metrics on the space $\mathcal{P}(\mathbb{R}^d)$ of measures on $\mathbb{R}^d$.

**Theorem 5.3.** *CircularTSW and SpatialTSW are metrics on the space $\mathcal{P}(\mathbb{R}^d)$.*

The proof of Theorem 5.3 is presented in Appendix B.5. The algorithms for CircularTSW and SpatialTSW are presented in Appendix D.1 (Alg. 1 and 2 respectively).

## 5.2. Components in CircularTSW and SpatialTSW

Both CircularTSW and SpatialTSW distances depend on the choice of splitting maps $\alpha$. Additionally, CircularTSW is influenced by the parameter $r \geq 0$, whereas SpatialTSW is determined by the selection of the injective map $h$.

**Splitting maps $\alpha$.** For both CircularTSW and SpatialTSW, we follow the construction of the splitting map $\alpha$ as proposed in (Tran et al., 2025b), defined as:

$$\alpha(x, \mathcal{L})_l = \mathrm{softmax}\Big(\{d(x, \mathcal{L})_i\}_{i=1}^k\Big), \tag{19}$$

where $d(x, \mathcal{L})_i$ represents the distance between $x$ and $i^{\text{th}}$ line of $\mathcal{L}$. This choice of $\alpha$ ensures the $E(d)$-invariance while also incorporating positional information between a point and the tree system. Consequently, it results in meaningful and varied mass distributions that adapt to each specific system. Moreover, the use of the softmax function guarantees that $\alpha$ produces a valid probability vector in the standard simplex $\Delta_{k-1}$. The splitting map for CircularTSW is further discussed in Appendix D.4.

**Radius $r$ in CircularTSW.** In the formulation of the original Circular Radon Transform, the radius $r$ plays a crucial role. Together with $\theta \in \mathbb{S}^{d-1}$, the term $r\theta$ spans the entire space $\mathbb{R}^d$. This ensures that the level sets defined by the defining circular function in Eq. (6) can represent arbitrary $(d-1)$-dimensional spheres in $\mathbb{R}^d$:

$$\{y \in \mathbb{R}^d : t = \|y - r\theta\|_2\} \text{ for } t \geq 0. \tag{20}$$

However, in the framework of Tree-Sliced methods, since the sources within tree systems already encompass the entire space $\mathbb{R}^d$, considering all values of $r \geq 0$ becomes redundant. In other words, for a fixed $r \geq 0$, the level sets arising in the Circular Radon Transform on Systems of Lines from Eq. (9) can still effectively represent arbitrary $(d-1)$-dimensional spheres in $\mathbb{R}^d$:

$$\{y \in \mathbb{R}^d : t = \|y - x_i - r\theta_i\|_2\} \text{ for } t \geq 0. \tag{21}$$

This is why, in the definition of CircularTSW, we fix a specific $r \geq 0$. Furthermore, we want to examine CircularTSW$_{r=0}$, a special case of CircularTSW when $r = 0$. In this scenario, by selecting the concurrent tree space as proposed in Tran et al. (2025b), the practical implementation of CircularTSW becomes significantly more

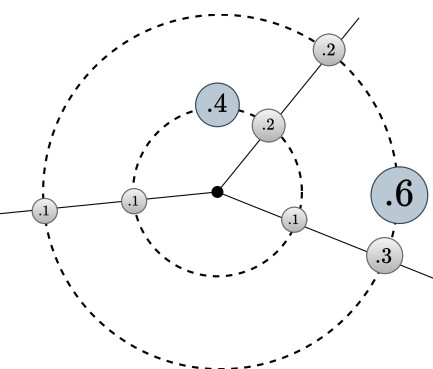

Figure 1: An illustration depicting CircularTSW$_{r=0}$. Given a support, the projection coordinates are identical when projected onto $k$ lines.

efficient, reducing computational complexity. In summary, transforming a measure $\mu \in \mathcal{P}(\mathbb{R}^d)$ involves projecting its support onto $k$ lines in the tree system. When $r = 0$, all support points of $\mu$ share the same coordinates when projected onto these $k$ lines. As a result, projecting and sorting cost is reduced. Figure 1 illustrates this phenomenon. Furthermore, we emphasize that CircularTSW$_{r=0}$ is specifically designed for tree settings, where splitting maps play a crucial role. Since the coordinates are identical across these lines, the tree structure and the distance-based splitting map are necessary to distinguish between the $k$ lines. Empirical results in Appendix D.5 show that CircularTSW$_{r=0}$ performs well in a tree setting but poorly in original sliced setting.

**The choice of the map $h$ in SpatialTSW.** As discussed in Section 4.1, the map $h$ in SpatialTSW must be both injective and continuous. One approach to selecting this map is based on odd degree homogeneous polynomials. However, following the constructions in (Kolouri et al., 2019; Rouvière, 2015) can lead to an excessively large new dimension $d_\theta = \binom{m+d-1}{d-1}$. To address this, we propose an alternative approach: Consider $h : \mathbb{R}^d \to \mathbb{R}^d$ defined by:

$$h(x_1, \ldots, x_d) = (f_1(x_1), \ldots, f_d(x_d)), \qquad (22)$$

where $\{f_i \colon \mathbb{R} \to \mathbb{R}\}_{i=1}^d$ are injective and continuous functions. A simple choice for $f_i$ is an odd-degree polynomial that remains injective, such as $f_i(x) = x_i + x_i^3$. Another approach, inspired by (Chen et al., 2022), involves concatenating the input with the output of a neural network by concatenating input with an arbitrary neural network $\phi(\cdot)$. Specifically, we define $h \colon \mathbb{R}^d \to \mathbb{R}^{d+d'}$ as $h(x) = (x, \phi(x))$, where $\phi \colon \mathbb{R}^d \to \mathbb{R}^{d'}$ is a neural network. This choice introduces learnable parameters for $h$, offering a trade-off between potentially improving performance and increasing computational cost.

### 5.3. Computational Complexity

Consider two discrete measures $\mu, \nu \in \mathcal{P}(\mathbb{R}^d)$ with $m, n$ support points, respectively. The computational complexity of the original Sliced Wasserstein distance is $\mathcal{O}(Ln \log n + Ldn)$, where $L$ represents the number of samples used in the Monte Carlo approximation (Peyré et al., 2019; Nguyen et al., 2024a). In comparison, the computational complexity of the Tree-Sliced Wasserstein (TSW) distances, such as TSW-SL in (Tran et al., 2025c) or Db-TSW in (Tran et al., 2025b), is $\mathcal{O}(Lkn \log n + Lkdn)$, where $L$ denotes the number of tree samples used in the Monte Carlo approximation, and $k$ is the number of lines per tree. This computational difference highlights why a fair comparison between the sliced method and the tree-sliced method requires ensuring that the total number of directions remains the same. For SpatialTSW, its computational complexity is $\mathcal{O}(Lkn \log n + Lkd_\theta n)$, with an additional initial cost for computing the function $h$. This complexity matches TSW-SL and Db-TSW but operates in the transformed space $\mathbb{R}^{d_\theta}$. For CircularTSW with a general $r \geqslant 0$, the complexity remains $\mathcal{O}(Lkn \log n + Lkd_\theta n)$, yet it achieves faster empirical runtime than Db-TSW by computing vector norms instead of vector products. Notably, for $r = 0$, CircularTSW$_{r=0}$ improves complexity to $\mathcal{O}(Ln \log n + Lkd_\theta n)$. This reduction arises because the $\mathcal{O}(n \log n)$ sorting step per line is required for only a single line, rather than all $k$ lines in a tree, as illustrated in Figure 1. The empirical efficiency of CircularTSW and CircularTSW$_{r=0}$ is demonstrated in Figure 2, where we use $L = 10000$ for SW and $L = 2500, k = 4$ for Tree-Sliced methods, following the practical setting used in the Diffusion Model experiment. CircularTSW$_{r=0}$ scales efficiently with the number of supports $n$ and is the only Tree-Sliced method that closely matches the speed of vanilla SW.

### 5.4. Spatial Spherical Tree-Sliced Wasserstein Distance

From the Spatial Spherical Radon Transform on Spherical Trees, we introduce a spherical variant of the Spatial Tree-Sliced Wasserstein distance, referred to as SpatialSTSW distance. A detailed derivation, along with the selection of the corresponding injective map and theoretical proofs for the SpatialSTSW distance, is provided in Appendix C.4.

## 6. Experimental Results

In this section, we thoroughly assess the validity of our proposed metric through extensive numerical experiments on both Euclidean and spherical datasets.

### 6.1. Euclidean Datasets

**Denoising Diffusion Generative Adversarial Network.** This experiment investigates training denoising diffusion

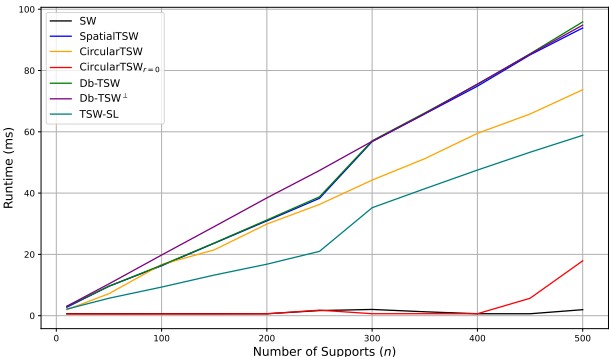

Figure 2: Runtime comparison of our proposed distances and baseline methods. We randomize the measures and projection directions and benchmark runtime over 10 runs. CircularTSW$_{r=0}$ is comparable to SW and significantly outperforms existing Tree-Sliced distances in terms of speed.

models for unconditional image synthesis. Following the approach of Nguyen et al. (2024b), we incorporate a Wasserstein distance into the Augmented Generalized Mini-batch Energy (AGME) loss function of the Denoising Diffusion Generative Adversarial Network (DDGAN) (Xiao et al., 2021). We benchmark our proposed methods – SpatialTSW-DD, CircularTSW, and CircularTSW$_{r=0}$ – against Sliced and Tree-Sliced Wasserstein-based DDGAN variants, as detailed in Table 1. All models are trained for 1800 epochs on the CIFAR10 dataset (Krizhevsky et al., 2009). For vanilla SW and its variants, we follow the parameter settings from Nguyen et al. (2024b), using $L = 10000$. For Tree-Sliced methods, including our own, we adopt the configuration from Tran et al. (2025b), setting $L = 2500$ and $k = 4$. Further details on this experiment can be found in Appendix D.6

The results in Table 1 show that SpatialTSW-DD, CircularTSW-DD, and CircularTSW$_{r=0}$-DD achieve notable improvements in FID compared to all baselines. They surpass the current state-of-the-art OT-based DDGAN, Db-TSW-DD$^\perp$ (Tran et al., 2025b), by margins of 0.01, 0.2, and 0.05, respectively. Additionally, our methods offer faster training times compared to existing Tree-Sliced approaches. CircularTSW-DD and CircularTSW reduce training time relative to Db-TSW-DD$^\perp$ by 10% and 19%, respectively. These enhancements in both training efficiency and model performance underscore the practical advantages of our proposed methods.

**Gradient Flow.** The goal of gradient flow is to minimize the distance between a source distribution $\mu$ and a target distribution $\nu$ through gradient-based optimization. The update rule follows $\partial_t \mu_t = -\nabla \mathcal{D}(\mu_t, \nu), \mu_0 = \mathcal{N}(0,1)$, where $\mu_t$ represents the distribution at time $t$, and $\nabla \mathcal{D}(\mu_t, \nu)$ is

Table 1: Fréchet Inception Distance (FID) scores and per-epoch training times of different DDGAN variants for unconditional generation on CIFAR-10.

| Model | FID ↓ | Time/Epoch(s) ↓ |
|---|---|---|
| DDGAN (Xiao et al., 2021) | 3.64 | 188 |
| SW-DD (Nguyen et al., 2024b) | 2.90 | 192 |
| DSW-DD (Nguyen et al., 2024b) | 2.88 | 1268 |
| EBSW-DD (Nguyen et al., 2024b) | 2.87 | 188 |
| RPSW-DD (Nguyen et al., 2024b) | 2.82 | 194 |
| IWRPSW-DD (Nguyen et al., 2024b) | 2.70 | 194 |
| TSW-SL-DD (Tran et al., 2025c) | 2.83 | 249 |
| Db-TSW-DD (Tran et al., 2025b) | 2.60 | 256 |
| Db-TSW-DD$^\perp$ (Tran et al., 2025b) | 2.53 | 262 |
| SpatialTSW-DD (ours) | 2.52 | 262 |
| CircularTSW-DD (ours) | **2.33** | 234 |
| CircularTSW$_{r=0}$-DD (ours) | 2.48 | 211 |

the gradient of the distance function $\mathcal{D}$ with respect to $\mu_t$. We evaluate SpatialTSW, CircularTSW, CircularTSW$_{r=0}$, and several established Sliced-Wasserstein (SW) variants, including vanilla SW (Bonneel et al., 2015), MaxSW (Deshpande et al., 2019), LCVSW (Nguyen & Ho, 2023), SWGG (Mahey et al., 2023), alongside the recently introduced Tree-Sliced distances, such as TSW-SL (Tran et al., 2025c), Db-TSW$^\perp$, and Db-TSW (Tran et al., 2025b). We conduct experiments on the *25 Gaussians* dataset and use the Wasserstein distance to evaluate the average distance between the source and target distributions. We report results over 5 runs at iterations 500, 1000, 1500, 2000, and 2500.

The results presented in Table 2 demonstrate that SpatialTSW achieves the best performance across all iterations, reaching a final $W_2$ distance of $1.17\mathrm{e}{-7}$ at the last step. This represents a significant improvement over vanilla SW ($3.59\mathrm{e}{-2}$) and LCVSW ($9.28\mathrm{e}{-3}$). Furthermore, compared to the best existing Tree-Sliced distances, SpatialTSW achieves better results than Db-TSW ($1.3\mathrm{e}{-7}$), exhibiting faster convergence, and maintaining similar computational efficiency. In this experiment, SpatialTSW outperforms CircularTSW, aligning with the findings from (Kolouri et al., 2019), where polynomial-based defining functions yield superior results over circular defining functions in gradient flow tasks. Nevertheless, both CircularTSW and CircularTSW$_{r=0}$ still achieve better results than vanilla SW while offering significant computational speedups. Specifically, CircularTSW and CircularTSW$_{r=0}$ are approximately 5% and 16% faster than vanilla SW, respectively.

### 6.2. Spherical Datasets

**Gradient Flow on The Sphere.** We now evaluate the ability to learn distributions by iteratively minimizing $d(\nu, \mu)$, where $d$ is a distance metric such as SSW (Bonet et al., 2022), S3W variants (Tran et al., 2024a) and STSW (Tran et al., 2025a). In line with previous works (Tran et al., 2024a; 2025a), we consider a mixture of 12 von Mises-Fisher distri-

Table 2: Average Wasserstein distance between source and target distributions of 5 runs on 25 Gaussians datasets. All methods use 100 projecting directions.

| Methods | Iteration | | | | | Time/Iter($s$) |
|---|---|---|---|---|---|---|
| | 500 | 1000 | 1500 | 2000 | 2500 | |
| SW | 4.21e-1 | 1.54e-1 | 7.72e-2 | 4.97e-2 | 3.59e-2 | 0.0018 |
| MaxSW | 5.23e-1 | 2.36e-1 | 1.23e-1 | 8.04e-2 | 6.76e-2 | 0.1020 |
| SWGG | 6.59e-1 | 3.62e-1 | 1.92e-1 | 9.07e-2 | 4.42e-2 | 0.0019 |
| LCVSW | 3.46e-1 | 6.96e-2 | 2.26e-2 | 1.31e-2 | 9.28e-3 | 0.0019 |
| TSW-SL | 3.49e-1 | 8.10e-2 | 1.06e-2 | 2.68e-3 | 3.16e-6 | 0.0019 |
| Db-TSW | 3.50e-1 | 8.12e-2 | 1.09e-2 | 1.77e-3 | 1.30e-7 | 0.0020 |
| Db-TSW$^\perp$ | 3.52e-1 | 7.69e-2 | 2.73e-2 | 2.56e-3 | 2.03e-6 | 0.0021 |
| SpatialTSW | **3.20e-1** | **3.44e-2** | **2.95e-3** | **3.97e-4** | **1.17e-7** | 0.0021 |
| CircularTSW | 4.28e-1 | 1.20e-1 | 3.48e-2 | 1.41e-2 | 7.86e-3 | 0.0017 |
| CircularTSW$_{r=0}$ | 4.32e-1 | 1.22e-1 | 3.41e-2 | 1.45e-2 | 8.94e-3 | 0.0015 |

Table 3: Average Log of the Wasserstein distance between source and target distributions over 5 runs on a mixture of 12 vMFs.

| Methods | Epoch | | | | | Time/Epoch($s$) |
|---|---|---|---|---|---|---|
| | 50 | 100 | 150 | 200 | 250 | |
| SSW | -2.4274 | -2.7893 | -2.9226 | -2.9882 | -3.0313 | 0.4323 |
| S3W | -2.0204 | -2.1920 | -2.2615 | -2.2699 | -2.2734 | 0.0151 |
| RI-S3W (1) | -2.1107 | -2.5163 | -2.7295 | -2.8568 | -2.9447 | 0.0182 |
| RI-S3W (5) | -2.4399 | -2.8273 | -3.0093 | -3.1234 | -3.2145 | 0.0503 |
| ARI-S3W (30) | -2.6508 | -3.0279 | -3.2405 | -3.4385 | -3.6661 | 0.1884 |
| STSW | **-2.9545** | **-3.5322** | -3.9992 | -4.3623 | -4.6486 | **0.0134** |
| SpatialSTSW | -2.8824 | -3.4626 | **-4.0903** | **-4.5368** | **-4.6859** | 0.0145 |

butions (vMFs) from which we have access to a sample set $\{y_i\}_{i=1}^{M}$ with $M = 2400$. The optimization procedure uses projected gradient descent (Bonet et al., 2022), applied on the sphere with full-batch size. Table 3 illustrates the evolution of the log 2-Wasserstein distance at epochs 50, 100, 150, 200, and 250, averaged over 5 runs. From these results, SpatialSTSW demonstrates better performance compared to baselines while maintaining computational efficiency close to STSW.

**Self-Supervised Learning (SSL).** In earlier work, Wang & Isola (2020) have shown that the contrastive objective can be broken down into an alignment loss, which ensures that representations of similar inputs are close together, and a uniformity loss, which prevents representations from collapsing by encouraging them to spread out evenly. Inspired by Bonet et al. (2022), we replace the Gaussian kernel in uniformity loss with our SpatialSTSW:

$$\mathcal{L} = \underbrace{\frac{1}{n}\sum_{i=1}^{n}\left\|z_i^A - z_i^B\right\|_2^2}_{\text{Alignment loss}} +$$
$$\underbrace{\frac{\lambda}{2}\left(\text{SpatialSTSW}(z^A,\nu) + \text{SpatialSTSW}(z^B,\nu)\right)}_{\text{Uniformity loss}},$$

where $\nu = \mathcal{U}(\mathbb{S}^d)$ is the uniform distribution on $\mathbb{S}^d$, $z^A, z^B \in \mathbb{R}^{n\times(d+1)}$ are feature embeddings of two augmented views of the same image and $\lambda > 0$ is regularization factor that balances the loss components. Follow-

Table 4: Accuracy of the linear classifier on encoded (E) features and projected (P) features on $\mathbb{S}^9$. ARI-S3W and RI-S3W use 5 rotations.

| Method | Acc. E(%) ↑ | Acc. P(%) ↑ | Time (s/ep.) |
|---|---|---|---|
| Hypersphere | 79.78 | 74.60 | 13.10 |
| SimCLR | 79.86 | 72.79 | **12.71** |
| SSW | 70.37 | 64.76 | 13.31 |
| S3W | 78.53 | 73.73 | 12.90 |
| RI-S3W (5) | 79.96 | 74.02 | 13.08 |
| ARI-S3W (5) | 80.06 | 75.10 | 13.01 |
| STSW | 80.51 | 76.79 | 12.81 |
| SpatialSTSW | **80.68** | **77.31** | 12.87 |

ing a similar approach to Bonet et al. (2022); Tran et al. (2024a; 2025a), we use the above objective function to pretrain a ResNet18 (He et al., 2016) encoder on CIFAR-10 (Krizhevsky et al., 2009) for 200 epochs and then train a linear classifier to evaluate learned features. The results in Table 4 indicate that SpatialSTSW achieves the best performance compared to various baseline methods, including Hypersphere (Wang & Isola, 2020), SimCLR (Chen et al., 2020), SSW, S3W variants, and STSW.

**Sliced-Wasserstein Auto-Encoder.** In this study, we utilize the Sliced-Wasserstein Auto-Encoder (SWAE) framework introduced by Kolouri et al. (2018) to evaluate the performance of various distances, including SW, SSW (Bonet et al., 2022), S3W variants (Tran et al., 2024a), and STSW (Tran et al., 2025a). The target of SWAE is to ensure that the encoded embeddings follow a predefined prior distribution $q$ in the latent space. Let the encoder be denoted as $\varphi : \mathcal{X} \to \mathbb{S}^d$ and the decoder as $\psi : \mathbb{S}^d \to \mathcal{X}$. The optimizing objective is defined as:

$$\min_{\varphi,\psi} \mathbb{E}_{x\sim p}\left[c(x, \psi(\varphi(x)))\right] + \lambda \cdot \text{SpatialSTSW}\left(\varphi_\sharp p, q\right),$$

where $p$ represents the data distribution, $\lambda$ acts as a regularization weight, and $c(\cdot, \cdot)$ measures the reconstruction error. For reconstruction loss, we use Binary Cross Entropy (BCE) and adopt a mixture of 10 von Mises-Fisher (vMF) distributions as the prior. We report results in Table 5, where we evaluate performance using the same metrics as in Tran et al. (2024a; 2025a). We observe that SpatialSTSW achieves better results in terms of $\log W_2$ and NLL while maintaining a competitive reconstruction loss (BCE) and efficient computation times.

## 7. Conclusion

This paper introduces the Circular Tree-Sliced Wasserstein Distance (CircularTSW) and the Spatial Tree-Sliced Wasserstein Distance (SpatialTSW) as novel approaches for comparing probability measures in Euclidean spaces. These

Table 5: CIFAR-10 results for SWAE evaluated on latent regularization.

| Method | $\log W_2 \downarrow$ | NLL $\downarrow$ | BCE $\downarrow$ | Time (s/ep.) |
|---|---|---|---|---|
| SW | -3.3181 | -0.0010 | 0.6330 | **6.8939** |
| SSW | -2.3425 | 0.0037 | 0.6316 | 16.8639 |
| S3W | -3.3181 | 0.0018 | **0.6307** | 8.9703 |
| RI-S3W | -3.1857 | -0.0034 | 0.6357 | 10.1904 |
| ARI-S3W | -3.3850 | 0.0020 | 0.6328 | 9.5882 |
| STSW | -3.4098 | -0.0045 | 0.6347 | 7.1623 |
| SpatialSTSW | **-3.4254** | **-0.0049** | 0.6368 | 7.2811 |

approaches integrate nonlinear projection techniques from the Sliced Wasserstein distance into the recent Tree-Sliced Wasserstein framework, resulting in enhanced performance and more efficient metric computations. The paper presents a formal derivation of these metrics and provides comprehensive theoretical guarantees to ensure their practical applicability. Furthermore, the proposed techniques are extended to the tree-sliced framework with a spherical setting. Experimental evaluations show that CircularTSW, SpatialTSW, and their spherical variant consistently outperform state-of-the-art Sliced Wasserstein and Tree-Sliced Wasserstein methods across various tasks, including gradient flows and diffusion models, while maintaining comparable or improved runtime efficiency. These results highlight Tree-Sliced Wasserstein distance as a promising and impactful research direction, complementing the Sliced Wasserstein distance in practical applications.

## Acknowledgements

We thank the area chairs and anonymous reviewers for their comments. TL gratefully acknowledges the support of JSPS KAKENHI Grant number 23K11243, and Mitsui Knowledge Industry Co., Ltd. grant.

## Impact Statement

This paper presents work whose goal is to advance the field of Machine Learning. There are many potential societal consequences of our work, none which we feel must be specifically highlighted here.

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

| | |
|---|---|
| $\mathbb{R}^d$ | $d$-dimensional Euclidean space |
| $\mathbb{S}^d$ | $d$-dimensional hypersphere |
| $\|\cdot\|_2$ | Euclidean norm |
| $\langle \cdot, \cdot \rangle$ | standard dot product |
| $\mathbb{S}^{d-1}$ | $(d-1)$-dimensional hypersphere |
| $\theta$ | unit vector |
| $\sqcup$ | disjoint union |
| $L^1(X)$ | space of Lebesgue integrable functions on $X$ |
| $\mathcal{P}(X)$ | space of probability distributions (or measures) on $X$ |
| $\mu, \nu$ | measures |
| $\delta(\cdot)$ | 1-dimensional Dirac delta function |
| $\mathcal{U}(\mathbb{S}^{d-1})$ | uniform distribution on $\mathbb{S}^{d-1}$ |
| $\sharp$ | pushforward (measure) |
| $\mathcal{C}(X,Y)$ | space of continuous maps from $X$ to $Y$ |
| $d(\cdot, \cdot)$ | metric in metric space |
| $\mathrm{T}(d)$ | translation group of order $d$ |
| $\mathrm{O}(d)$ | orthogonal group of order $d$ |
| $\mathrm{E}(d)$ | Euclidean group of order $d$ |
| $g$ | element of group |
| $\mathrm{W}_p$ | $p$-Wasserstein distance |
| $\mathrm{SW}_p$ | Sliced $p$-Wasserstein distance |
| $\mathcal{L}$ | system of lines, tree system |
| $r_y^x$ | spherical ray |
| $\mathcal{T}, \mathcal{T}_{y_1,\ldots,y_k}^x$ | spherical tree |
| $\mathbb{L}_k^d$ | space of symtems of $k$ lines in $\mathbb{R}^d$ |
| $L$ | number of tree systems |
| $k$ | number of lines in a system of lines or a tree system |
| $\mathcal{R}$ | original Radon Transform |
| $\mathcal{R}^\alpha$ | Radon Transform on Systems of Lines, or Radon Transform on Spherical Trees |
| $\Delta_{k-1}$ | $(k-1)$-dimensional standard simplex |
| $\alpha$ | splitting map |
| $\delta$ | Dirac delta function |
| $\mathbb{T}$ | space of tree systems |
| $\sigma$ | distribution on space of tree systems |

# Supplemental Material for "Tree-Sliced Wasserstein Distance with Nonlinear Projection"

The supplementary is organized into four parts as follows:

- In Section A, we provide background for Tree-Sliced Wasserstein distance.

- In Section B, we derive theoretical proofs for Radon transform on systems of lines with nonlinear projection.

- In Section C, we describe Radon transform on systems of lines for spherical functions.

- In Section D, we provide further details for the experiments.

## A. Background for Tree-Sliced Wasserstein Distance

In this section, we briefly outline the notion of the Radon Transform on Systems of Lines (Tran et al., 2025c) with its distance-based extension (Tran et al., 2025b).

**Building blocks of Tree-sliced Wasserstein distance on Systems of Lines.**   The Tree-sliced Wasserstein distance on Systems of Lines is constructed step-by-step as follows:

1. Given a positive number $d$ presenting the dimension.

2. A *line* in $\mathbb{R}^d$ is an element $l = (x, \theta) \in \mathbb{R}^d \times \mathbb{S}^{d-1}$. Here, $x$ is called the *source* and $\theta$ is called the *direction* of the line.

3. A *system of $k$ lines* in $\mathbb{R}^d$ is an element of $(\mathbb{R}^d \times \mathbb{S}^{d-1})^k$. Denote a system of lines as $\mathcal{L}$, and the space of all systems of $k$ lines by $\mathbb{L}_k^d$.

4. A *point $x$ in $\mathcal{L}$* can be parameterized as $x_i + t \cdot \theta_i$, where $i$ is the index of the line, and $t$ is the coordinate of the point on that $i^{\text{th}}$ lines.

5. A system of line $\mathcal{L}$ with additional tree structure is called a *tree system* (see (Tran et al., 2025c;b)). Each tree system is a measure space, endowed with a tree metric.

6. A *space of trees* (collections of all tree systems with the same tree structure) is denoted by $\mathbb{T}$ with a probability distribution $\sigma$ on $\mathbb{T}$, which comes from the tree sampling process.

7. For $\mathcal{L} \in \mathbb{L}_k^d$, *the space of integrable functions on $\mathcal{L}$* is:

$$L^1(\mathcal{L}) = \left\{ f \colon \mathcal{L} \to \mathbb{R} \ : \ \|f\|_{\mathcal{L}} = \sum_{i=1}^{k} \int_{\mathbb{R}} |f(x_i + t \cdot \theta_i)| \, dt < \infty \right\}. \tag{23}$$

8. A splitting map $\alpha$ is a continuous map from $\mathbb{R}^d \times \mathbb{L}_k^d$ to the $(k-1)$-dimensional standard simplex $\Delta_{k-1}$, i.e. $\alpha \in \mathcal{C}(\mathbb{R}^d \times \mathbb{L}_{;}^d \Delta_{k-1})$. For $f \in L^1(\mathbb{R}^d)$, we define:

$$\mathcal{R}_{\mathcal{L}}^\alpha f \ : \quad \mathcal{L} \quad \longrightarrow \quad \mathbb{R} \tag{24}$$

$$x_i + t \cdot \theta_i \longmapsto \int_{\mathbb{R}^d} f(y) \cdot \alpha(y, \mathcal{L})_i \cdot \delta\left(t - \langle y - x_i, \theta_i \rangle\right) \, dy. \tag{25}$$

The function $\mathcal{R}_{\mathcal{L}}^\alpha f$ is in $L^1(\mathcal{L})$.

9. The operator:

$$\mathcal{R}^\alpha \ : \ L^1(\mathbb{R}^d) \ \longrightarrow \ \prod_{\mathcal{L} \in \mathbb{L}_k^d} L^1(\mathcal{L})$$

$$f \ \longmapsto \ (\mathcal{R}_{\mathcal{L}}^\alpha f)_{\mathcal{L} \in \mathbb{L}_k^d}$$

is called the *Radon Transform on Systems of Lines*.

10. When the splitting map $\alpha$ is $\mathrm{E}(d)$-invariant (this $\mathrm{E}(d)$-invariance will be described in the next part), the Radon Transform on Systems of Lines is *injective* (see (Tran et al., 2025b)).

11. The *Tree-Sliced Wasserstein Distance on Systems of Lines*, denoted by TSW-SL as in (Tran et al., 2025c), or its Distance-based variant Db-TSW as in (Tran et al., 2025b), between $\mu, \nu$ in $\mathcal{P}(\mathbb{R}^d)$ is defined by:

$$\text{Db-TSW}(\mu, \nu) = \int_{\mathbb{T}_k^d} \mathrm{W}_{d_{\mathcal{L}}, 1}(\mathcal{R}_{\mathcal{L}}^{\alpha} f_{\mu}, \mathcal{R}_{\mathcal{L}}^{\alpha} f_{\nu}) \, d\sigma(\mathcal{L}). \tag{26}$$

We choose the notation Db-TSW since this variant is a generalization of TSW-SL. Db-TSW is identical with the definition of SW when $k = 1$, i.e. tree systems in $\mathbb{L}_k^d$ have only one line.

12. The Db-TSW distance is a metric on $\mathcal{P}(\mathbb{R}^d)$ (see (Tran et al., 2025b)).

13. It is worth noting that, on tree systems, optimal transport problems admits closed-form expression, since it is a metric space with tree metric (see (Le et al., 2019)). Leveraging this closed-form expression and the Monte Carlo method, the distance in Eq. (26) can be efficiently approximated by a closed-form expression. Additionally, for the $p$-order Wasserstein with $p > 1$, one may consider the scalable variant—Sobolev transport (Le et al., 2022), which also yields a closed-form expression for a fast computation, and generalizes tree-Wasserstein (i.e., 1-order Wasserstein on a tree) to a more general settings such as for $p > 1$, and for measures on a graph.

**The group** $\mathrm{E}(d)$ **and its action.** The Euclidean group E(d) is the group of all transformations of $\mathbb{R}^d$ that preserve the Euclidean distance between any two points. It is the semidirect product between $\mathrm{T}(d)$ and $\mathrm{O}(d)$, i.e.

$$\mathrm{E}(d) \simeq \mathrm{T}(d) \rtimes \mathrm{O}(d), \tag{27}$$

where $\mathrm{T}(d)$ is group of all translations in $\mathbb{R}^d$ and $\mathrm{O}(d)$ is the orthogonal group of $\mathbb{R}^d$. Each element $g$ of $\mathrm{T}(d)$ can be presented as a pair:

$$g = (Q, a) \in \mathrm{T}(d) \quad \text{where} \quad Q \in \mathrm{O}(d) \text{ and } a \in \mathbb{R}^d. \tag{28}$$

The group E(d) acts on $\mathbb{R}^d$ naturally as follows: For $x \in \mathbb{R}^d$ and $g = (Q, a) \in \mathrm{T}(d)$, we have:

$$(g, x) \longmapsto gx = Q \cdot x + a. \tag{29}$$

It naturally induces a group action on the set of all lines in $\mathbb{R}^d$, i.e. $\mathbb{R}^d \times \mathbb{S}^{d-1}$: For $l = (x, \theta) \in \mathbb{R}^d \times \mathbb{S}^{d-1}$ and $g = (Q, a) \in \mathrm{E}(d)$, we have:

$$(g, l) \longmapsto gl = (Q \cdot x + a, Q \cdot \theta) \in \mathbb{R}^d \times \mathbb{S}^{d-1}. \tag{30}$$

For $\mathcal{L} = \left\{ l_i = (x_i, \theta_i) \right\}_{i=1}^{k} \in \mathbb{L}_k^d$, the action of $\mathrm{E}(d)$ on $\mathbb{L}_k^d$ is defined as:

$$g\mathcal{L} = \left\{ gl_i = (Q \cdot x_i + a, Q \cdot \theta_i) \right\}_{i=1}^{k} \in \mathbb{L}_k^d. \tag{31}$$

The tree structure of a tree system is preserved under the action of $\mathrm{E}(d)$ (see (Tran et al., 2025c;b)). In other words, if $\mathcal{L} \in \mathbb{T}$ is a tree system, then $g\mathcal{L}$ is also a tree system. The group action of $\mathrm{E}(d)$ on $\mathbb{L}_k^d$ induces a group action of $\mathrm{E}(d)$ on $\mathbb{T}$.

$\mathrm{E}(d)$-**invariant splitting maps.** A splitting map $\alpha \in \mathcal{C}(\mathbb{R}^d \times \mathbb{L}_k^d, \Delta_{k-1})$ is $\mathrm{E}(d)$-invariant, if:

$$\alpha(gy, g\mathcal{L}) = \alpha(y, \mathcal{L}), \tag{32}$$

for all $(y, \mathcal{L}) \in \mathbb{R}^d \times \mathbb{L}_k^d$ and $g \in \mathrm{E}(d)$.

*Remark* A.1. Equivariance is widely used in machine learning across various contexts, including equivariant models (Tran et al., 2024c) and equivariant metanetworks (Vo et al., 2024; Tran et al., 2024d;b;d). These approaches leverage symmetries in data or model architectures to improve generalization, reduce sample complexity, and ensure consistency under group transformations.

# B. Theoretical Proofs for Radon Transform on Systems of Lines with Nonlinear Projection

## B.1. $\mathcal{CR}^{\alpha}$ is well-defined

We show that the Circular Radon Transform on Systems of Lines is well-defined.

*Proof.* Recall from Eq. (9), we have:

$$\mathcal{CR}^{\alpha} \colon L^1(\mathbb{R}^d) \longrightarrow \prod_{\mathcal{L} \in \mathbb{L}_k^d, r \geqslant 0} L^1(\mathcal{L}) \quad \text{where} \quad f \longmapsto \left(\mathcal{CR}^{\alpha}_{\mathcal{L},r} f\right)_{\mathcal{L} \in \mathbb{L}_k^d, r \geqslant 0}, \tag{33}$$

$$\text{and} \quad \mathcal{CR}^{\alpha}_{\mathcal{L},r} f(x_i + t \cdot \theta_i) = \int_{\mathbb{R}^d} f(y) \cdot \alpha(y, \mathcal{L})_i \cdot \delta\left(t - \|y - x_i - r\theta_i\|_2\right) \, dy. \tag{34}$$

We have:

$$
\begin{aligned}
\|\mathcal{CR}^{\alpha}_{\mathcal{L},r} f\|_{\mathcal{L}} &= \sum_{i=1}^{k} \int_{\mathbb{R}} \left| \mathcal{CR}^{\alpha}_{\mathcal{L},r} f(x_i + t \cdot \theta) \right| \, dt_x \\
&= \sum_{i=1}^{k} \int_{\mathbb{R}} \left| \int_{\mathbb{R}^d} f(y) \cdot \alpha(y, \mathcal{L})_i \cdot \delta\left(t - \|y - x_i - r\theta_i\|_2\right) \, dy \right| \, dt \\
&\leqslant \sum_{i=1}^{k} \int_{\mathbb{R}} \left( \int_{\mathbb{R}^d} |f(y)| \cdot \alpha(y, \mathcal{L})_i \cdot \delta\left(t - \|y - x_i - r\theta_i\|_2\right) \, dy \right) \, dt \\
&= \sum_{i=1}^{k} \int_{\mathbb{R}^d} \left( \int_{\mathbb{R}} |f(y)| \cdot \alpha(y, \mathcal{L})_i \cdot \delta\left(t - \|y - x_i - r\theta_i\|_2\right) \, dt \right) \, dy \\
&= \sum_{i=1}^{k} \int_{\mathbb{R}^d} |f(y)| \cdot \alpha(y, \mathcal{L})_i \cdot \left( \int_{\mathbb{R}} \delta\left(t - \|y - x_i - r\theta_i\|_2\right) \, dt \right) \, dy \\
&= \sum_{i=1}^{k} \int_{\mathbb{R}^d} |f(y)| \cdot \alpha(y, \mathcal{L})_i \, dy \\
&= \int_{\mathbb{R}^d} |f(y)| \cdot \left( \sum_{i=1}^{k} \alpha(y, \mathcal{L})_i \right) \, dy \\
&= \int_{\mathbb{R}^d} |f(y)| \, dy \\
&= \|f\|_1. 
\end{aligned}
\tag{35}
$$

So $\mathcal{CR}^{\alpha}_{\mathcal{L},r} f \in L^1(\mathcal{L})$. It implies that the operator $\mathcal{CR}^{\alpha}_{\mathcal{L},r} \colon L^1(\mathbb{R}^d) \to L^1(\mathcal{L})$ is well-defined, as well as $\mathcal{CR}^{\alpha}$. $\qquad \square$

*Remark* B.1. Note that, from the above proof, we see that if $f \in \mathcal{P}(\mathbb{R}^d)$, i.e. $f \in L^1(\mathbb{R}^d)$, $\|f\|_1 = 1$ and $f(y) \geqslant 0$ for all $y \in \mathbb{R}^d$, we also have $\|\mathcal{CR}^{\alpha}_{\mathcal{L},r} f\|_{\mathcal{L}} = 1$ and $\mathcal{CR}^{\alpha}_{\mathcal{L},r} f(x_i + t \cdot \theta_i) \geqslant 0$ for all $x_i + t \cdot \theta_i \in \mathcal{L}$. It implies that $\mathcal{CR}^{\alpha}_{\mathcal{L},r} f \in \mathcal{P}(\mathcal{L})$.

## B.2. $\mathcal{H}^{\alpha}$ is well-defined

We show that the Spatial Radon Transform on Systems of Lines is well-defined.

*Proof.* Recall from Eq. (11), we have:

$$\mathcal{H}^{\alpha} \colon L^1(\mathbb{R}^d) \to \prod_{\mathcal{L} \in \mathbb{L}_k^{d_\theta}} L^1(\mathcal{L}) \quad \text{where} \quad f \mapsto \left(\mathcal{H}^{\alpha}_{\mathcal{L}} f\right)_{\mathcal{L} \in \mathbb{L}_k^{d_\theta}},$$

$$\text{and} \quad \mathcal{H}^{\alpha}_{\mathcal{L}} f(x_i + t \cdot \theta_i) = \int_{\mathbb{R}^d} f(y) \cdot \alpha(h(y), \mathcal{L})_i \cdot \delta\left(t - \langle h(y) - x_i, \theta_i \rangle\right) \, dy. \tag{36}$$

We have:

$$
\begin{aligned}
\|\mathcal{H}_{\mathcal{L}}^{\alpha} f\|_{\mathcal{L}} &= \sum_{i=1}^{k} \int_{\mathbb{R}} |\mathcal{H}_{\mathcal{L}}^{\alpha} f(x_i + t \cdot \theta_i)| \ dt_x \\
&= \sum_{i=1}^{k} \int_{\mathbb{R}} \left| \int_{\mathbb{R}^d} f(y) \cdot \alpha(h(y), \mathcal{L})_i \cdot \delta\left(t - \langle h(y) - x_i, \theta_i \rangle\right) \ dy \right| dt \\
&\leqslant \sum_{i=1}^{k} \int_{\mathbb{R}} \left( \int_{\mathbb{R}^d} |f(y)| \cdot \alpha(h(y), \mathcal{L})_i \cdot \delta\left(t - \langle h(y) - x_i, \theta_i \rangle\right) \ dy \right) dt \\
&= \sum_{i=1}^{k} \int_{\mathbb{R}^d} \left( \int_{\mathbb{R}} |f(y)| \cdot \alpha(h(y), \mathcal{L})_i \cdot \delta\left(t - \langle h(y) - x_i, \theta_i \rangle\right) \ dt \right) dy \\
&= \sum_{i=1}^{k} \int_{\mathbb{R}^d} |f(y)| \cdot \alpha(h(y), \mathcal{L})_i \cdot \left( \int_{\mathbb{R}} \delta\left(t - \langle h(y) - x_i, \theta_i \rangle\right) \ dt \right) dy \\
&= \sum_{i=1}^{k} \int_{\mathbb{R}^d} |f(y)| \cdot \alpha(h(y), \mathcal{L})_i \ dy \\
&= \int_{\mathbb{R}^d} |f(y)| \cdot \left( \sum_{i=1}^{k} \alpha(h(y), \mathcal{L})_i \right) dy \\
&= \int_{\mathbb{R}^d} |f(y)| \ dy \\
&= \|f\|_1.
\end{aligned}
\tag{37}
$$

So $\mathcal{H}_{\mathcal{L}}^{\alpha} f \in L^1(\mathcal{L})$. It implies the operator $\mathcal{H}_{\mathcal{L}}^{\alpha} \colon L^1(\mathbb{R}^d) \to L^1(\mathcal{L})$ is well-defined, as well as $\mathcal{H}^{\alpha}$. $\qquad\square$

*Remark* B.2. Note that, from the above proof, we see that if $f \in \mathcal{P}(\mathbb{R}^d)$, i.e. $f \in L^1(\mathbb{R}^d)$, $\|f\|_1 = 1$ and $f(y) \geqslant 0$ for all $y \in \mathbb{R}^d$, we also have $\|\mathcal{H}_{\mathcal{L}}^{\alpha} f\|_{\mathcal{L}} = 1$ and $\mathcal{H}_{\mathcal{L}}^{\alpha} f(x_i + t \cdot \theta_i) \geqslant 0$ for all $x_i + t \cdot \theta_i \in \mathcal{L}$. It implies that $\mathcal{H}_{\mathcal{L}}^{\alpha} f \in \mathcal{P}(\mathcal{L})$.

### B.3. Proof for Theorem 4.2

Recall the original Circular Radon Transform $\mathcal{CR}$ (Kuchment, 2006; Kolouri et al., 2019) as follows:

$$
\mathcal{CR} \colon \quad L^1(\mathbb{R}^d) \quad \longrightarrow \quad L^1(\mathbb{R} \times \mathbb{S}^{d-1} \times \mathbb{R}_{\geqslant 0})
$$
$$
f \quad \longmapsto \quad \mathcal{CR}f,
\tag{38}
$$

where:

$$
\mathcal{CR}f \colon \quad \mathbb{R} \times \mathbb{S}^{d-1} \times \mathbb{R}_{\geqslant 0} \quad \longrightarrow \quad \mathbb{R}
\tag{39}
$$
$$
(t, \theta, r) \quad \longmapsto \quad \int_{\mathbb{R}^d} f(y) \cdot \delta\left(t - \|y - r\theta\|_2\right) \ dy.
\tag{40}
$$

In (Kuchment, 2006), it is showed that the Circular Radon Transform $\mathcal{CR}$ is injective. We will leverage this result to prove the injectivity of the proposed Circular Radon Transform on Systems of Lines $\mathcal{CR}^{\alpha}$.

First, for each $\theta \in \mathbb{S}^{d-1}$, consider the tree system $\mathcal{L}^{(i)}$ consists of $k$ identical lines $(0, \theta)$. Define the function $g$ as follows:

$$
g \colon \quad \mathbb{R} \times \mathbb{S}^{d-1} \times \mathbb{R}_{\geqslant 0} \quad \longrightarrow \quad \mathbb{R}
\tag{41}
$$
$$
(t, \theta, r) \quad \longmapsto \quad \sum_{j=1}^{k} \mathcal{CR}_{\mathcal{L}^{(j)}, r}^{\alpha} f(x_{\mathcal{L}^{(j)}:i} + t \cdot \theta_{\mathcal{L}^{(j)}:i}).
\tag{42}
$$

Since

$$
\mathcal{CR}_{\mathcal{L}, r}^{\alpha} f(x_i + t \cdot \theta_i) = \int_{\mathbb{R}^d} f(y) \cdot \alpha(y, \mathcal{L})_i \cdot \delta\left(t - \|y - x_i - r\theta_i\|_2\right) \ dy.
\tag{43}
$$

We have:

$$g(t, \theta, r) = \sum_{j=1}^{k} \mathcal{CR}_{\mathcal{L}^{(j)}, r}^{\alpha} f(x_{\mathcal{L}^{(j)}:i} + t \cdot \theta_{\mathcal{L}^{(j)}:i}) \tag{44}$$

$$= \sum_{i=1}^{k} \int_{\mathbb{R}^d} f(y) \cdot \alpha(y, \mathcal{L}^{(j)})_i \cdot \delta\left(t - \|y - x_{\mathcal{L}^{(j)}:i} - r\theta_{\mathcal{L}^{(j)}:i}\|_2\right) \, dy \tag{45}$$

$$= \sum_{i=1}^{k} \int_{\mathbb{R}^d} f(y) \cdot \alpha(y, \mathcal{L}^{(j)})_i \cdot \delta\left(t - \|y - r\theta\|_2\right) \, dy \tag{46}$$

$$= \sum_{i=1}^{k} \int_{\mathbb{R}^d} f(y) \cdot \delta\left(t - \|y - r\theta\|_2\right) \cdot \left(\sum_{i=1}^{k} \alpha(y, \mathcal{L}^{(j)})_i\right) \, dy \tag{47}$$

$$= \sum_{i=1}^{k} \int_{\mathbb{R}^d} f(y) \cdot \delta\left(t - \|y - r\theta\|_2\right) \cdot \, dy \tag{48}$$

$$= \mathcal{CR} f. \tag{49}$$

It is clear that $\mathcal{CR}^{\alpha}$ is a linear operator. To prove $\mathcal{CR}^{\alpha}$ injective, consider $f \in \mathrm{Ker}(\mathcal{CR}^{\alpha})$. By the definition of $g$ in Eq. (41), we have $g$ is the function 0. But $g$ is exactly is the Circular Radon Transform of $f$, and since the Circular Radon Transform is injective, we conclude that $f$ is the function 0. In conclusion, $\mathcal{CR}^{\alpha}$ is injective.

### B.4. Proof for Theorem 4.3

We present the proof for Theorem 4.3.

*Proof.* Recall the Radon Transform on Systems of Lines $\mathcal{R}^{\alpha}$ (Tran et al., 2025b) as follows:

$$\mathcal{R}^{\alpha} : L^1(\mathbb{R}^d) \longrightarrow \prod_{\mathcal{L} \in \mathbb{L}_k^d} L^1(\mathcal{L})$$

$$f \longmapsto (\mathcal{R}_{\mathcal{L}}^{\alpha} f)_{\mathcal{L} \in \mathbb{L}_k^d}, \tag{50}$$

where

$$\mathcal{R}_{\mathcal{L}}^{\alpha} f : \qquad \mathcal{L} \longrightarrow \mathbb{R}$$

$$x_i + t \cdot \theta_i \longmapsto \int_{\mathbb{R}^d} f(y) \cdot \alpha(y, \mathcal{L})_i \cdot \delta\left(t - \langle y - x_i, \theta_i \rangle\right) \, dy, \tag{51}$$

It is proved in (Tran et al., 2025b) that $\mathcal{R}^{\alpha}$ is injective for $\mathrm{E}(d)$-invariant splitting map $\alpha$. We leverage this result to prove the Spatial Radon Transform on Systems of Lines is injective. First, by the injective continuous map $h : \mathbb{R}^d \to \mathbb{R}^{d_\theta}$, we show that the push-forward of $f \in \mathbb{R}^d$ via $h$, defined as:

$$h_\sharp f(y) = \begin{cases} f(h^{-1}(y)) & , \text{ for all } y \in \mathbb{R}^{d_\theta} \text{ such that } y \in h(\mathbb{R}^d), \\ 0 & , \text{ for all } y \in \mathbb{R}^{d_\theta} \text{ such that } y \notin h(\mathbb{R}^d). \end{cases} \tag{52}$$

has its Radon Transform on Systems of Lines, i.e. $\{\mathcal{R}_{\mathcal{L}}^{\alpha}(h_\sharp f)\}_{\mathcal{L} \in \mathbb{L}_k^{d_\theta}}$, equal to the Spatial Radon Transform on Systems of Lines of $f$, i.e. $\{\mathcal{H}_{\mathcal{L}}^{\alpha} f\}_{\mathcal{L} \in \mathbb{L}_k^{d_\theta}}$. In other words, for all $\mathcal{L} \in \mathbb{L}_k^{d_\theta}$, we have:

$$\mathcal{H}_{\mathcal{L}}^{\alpha} f = \mathcal{R}_{\mathcal{L}}^{\alpha}(h_\sharp f). \tag{53}$$

Indeed, we have:

$$\mathcal{R}_{\mathcal{L}}^{\alpha}(h_\sharp f)(x_i + t \cdot \theta_i) = \int_{\mathbb{R}^{d_\theta}} h_\sharp f(y) \cdot \alpha(y, \mathcal{L})_l \cdot \delta\left(t - \langle y - x_i, \theta_i \rangle\right) \, dy$$

$$
\begin{aligned}
&= \int_{h(\mathbb{R}^d)} h_\sharp f(y) \cdot \alpha(y, \mathcal{L})_l \cdot \delta\left(t - \langle y - x_i, \theta_i \rangle\right) \, dy \\
&= \int_{\mathbb{R}^d} f(h^{-1}(h(y))) \cdot \alpha(h(y), \mathcal{L})_l \cdot \delta\left(t - \langle h(y) - x_i, \theta_i \rangle\right) \, dy \\
&= \int_{\mathbb{R}^d} f(y) \cdot \alpha(h(y), \mathcal{L})_l \cdot \delta\left(t - \langle h(y) - x_i, \theta_i \rangle\right) \, dy \\
&= \mathcal{H}_{\mathcal{L}}^{\alpha} f(x_i + t \cdot \theta_i).
\end{aligned}
\tag{54}
$$

It is clear that $\mathcal{H}^\alpha$ is a linear operator. To prove $\mathcal{H}^\alpha$ is injective, consider $f \in \mathrm{Ker}(\mathcal{H}^\alpha)$. Since $\mathcal{H}_{\mathcal{L}}^\alpha f = \mathcal{R}_{\mathcal{L}}^\alpha(h_\sharp f)$, it implies that $h_\sharp f \in \mathrm{Ker}(\mathcal{R}^\alpha)$. Since $\mathcal{R}^\alpha$ is injective, it implies that $h_\sharp f$ is the function 0. By the definition of the push-forward $h_\sharp f$ as in Eq. (52), we conclude that $f$ is the function 0. In conclusion, $\mathcal{H}^\alpha$ is injective. $\qquad\square$

### B.5. Proof of Theorem 5.3

We show that CircularTSW is a metric on $\mathcal{P}(\mathbb{R}^d)$. The proof for SpatialTSW is similar.

*Proof.* We will show that:

$$
\mathrm{CircularTSW}(\mu, \nu) = \int_{\mathbb{T}_k^d} \mathrm{W}(\mathcal{CR}_{\mathcal{L},r}^\alpha f_\mu, \mathcal{CR}_{\mathcal{L},r}^\alpha f_\nu) \, d\sigma(\mathcal{L}),
\tag{55}
$$

is a metric on $\mathcal{P}(\mathbb{R}^d)$, by verifying its positive definiteness, symmetry and triangle inequality.

**Positive definiteness.** For $\mu, \nu \in \mathcal{P}(\mathbb{R}^d)$, it is clear that

$$
\mathrm{CircularTSW}(\mu, \mu) = 0,
\tag{56}
$$

and

$$
\mathrm{CircularTSW}(\mu, \nu) \geqslant 0.
\tag{57}
$$

If $\mathrm{CircularTSW}(\mu, \nu) = 0$, then $\mathrm{W}(\mathcal{CR}_{\mathcal{L},r}^\alpha f_\mu, \mathcal{CR}_{\mathcal{L},r}^\alpha f_\nu) = 0$ for almost every $\mathcal{L} \in \mathbb{T}_k^d$. Since W is a metric on $\mathcal{P}(\mathcal{L})$, we have $\mathcal{CR}_{\mathcal{L},r}^\alpha f_\mu = \mathcal{CR}_{\mathcal{L},r}^\alpha f_\nu$ for almost every $\mathcal{L} \in \mathbb{T}$. By Theorem 4.2, it implies that $\mu = \nu$.

**Symmetry.** For $\mu, \nu \in \mathcal{P}(\mathbb{R}^d)$, we have:

$$
\begin{aligned}
\mathrm{CircularTSW}(\mu, \nu) &= \int_{\mathbb{T}_k^d} \mathrm{W}(\mathcal{CR}_{\mathcal{L},r}^\alpha f_\mu, \mathcal{CR}_{\mathcal{L},r}^\alpha f_\nu) \, d\sigma(\mathcal{L}) \\
&= \int_{\mathbb{T}_k^d} \mathrm{W}(\mathcal{CR}_{\mathcal{L},r}^\alpha f_\nu, \mathcal{CR}_{\mathcal{L},r}^\alpha f_\mu) \, d\sigma(\mathcal{L}) \\
&= \mathrm{CircularTSW}(\nu, \mu)
\end{aligned}
\tag{58}
$$

So $\mathrm{CircularTSW}(\mu, \nu) = \mathrm{CircularTSW}(\nu, \mu)$.

**Triangle inequality.** For $\mu_1, \mu_2, \mu_3 \in \mathcal{P}(\mathbb{R}^D)$, we have:

$$
\begin{aligned}
&\mathrm{CircularTSW}(\mu_1, \mu_2) + \mathrm{CircularTSW}(\mu_2, \mu_3) \\
&= \int_{\mathbb{T}_k^d} \mathrm{W}(\mathcal{CR}_{\mathcal{L},r}^\alpha f_{\mu_1}, \mathcal{CR}_{\mathcal{L},r}^\alpha f_{\mu_2}) \, d\sigma(\mathcal{L}) + \int_{\mathbb{T}_k^d} \mathrm{W}(\mathcal{CR}_{\mathcal{L},r}^\alpha f_{\mu_2}, \mathcal{CR}_{\mathcal{L},r}^\alpha f_{\mu_3}) \, d\sigma(\mathcal{L}) \\
&= \int_{\mathbb{T}_k^d} \left( \mathrm{W}(\mathcal{CR}_{\mathcal{L},r}^\alpha f_{\mu_1}, \mathcal{CR}_{\mathcal{L},r}^\alpha f_{\mu_2}) + \mathrm{W}(\mathcal{CR}_{\mathcal{L},r}^\alpha f_{\mu_1}, \mathcal{CR}_{\mathcal{L},r}^\alpha f_{\mu_2}) \right) \, d\sigma(\mathcal{L}) \\
&\geqslant \int_{\mathbb{T}_k^d} \mathrm{W}(\mathcal{CR}_{\mathcal{L},r}^\alpha f_{\mu_1}, \mathcal{CR}_{\mathcal{T},r}^\alpha f_{\mu_3}) \, d\sigma(\mathcal{L}) \\
&= \mathrm{CircularTSW}(\mu_1, \mu_3).
\end{aligned}
\tag{59}
$$

The triangle inequality holds for CircularTSW.

In conclusion, CircularTSW is a metric on the space $\mathcal{P}(\mathbb{R}^d)$. $\qquad\square$

# C. Radon Transform on Systems of Lines for Spherical Functions

In this section, we review (Tran et al., 2025a) which proposes Spherical Radon Transform on Spherical Trees and Spherical Tree-Sliced Wasserstein distance, which are analogs to Radon Transform on Systems of Lines (Tran et al., 2025c;b) and corresponding metric, applied for spherical functions. Then, we explain how to apply Generalized-like framework in this paper for spherical settings

## C.1. Background for Spherical Radon Transform on Spherical Trees

To make this easy to follow, we will follow the construction of Appendix A. **Building blocks of Spherical Tree-Sliced Wasserstein distance.**

1. Given a positive number $d$ presenting the dimension. We will work with functions on the d-dimensional hypersphere $\mathbb{S}^d \subset \mathbb{R}^{d+1}$, where:

$$\mathbb{S}^d := \left\{ x = (x_0, x_1, \ldots, x_d) \in \mathbb{R}^{d+1} \ : \ \|x\|_2 = 1 \right\} \subset \mathbb{R}^{d+1}.$$

   Note that $\mathbb{S}^d$ is a metric space with the metric $d_{\mathbb{S}^d}$ defined as $d_{\mathbb{S}^d}(a, b) = \arccos \langle a, b \rangle_{\mathbb{R}^{d+1}}$.

2. The stereographic projection corresponding to $x \in \mathbb{S}^d$ is defined by:

$$\begin{aligned}
\varphi_x : \quad & \mathbb{S}^d \setminus \{x\} \longrightarrow H_x \\
& y \longmapsto \frac{-\langle x, y \rangle}{1 - \langle x, y \rangle} \cdot x + \frac{1}{1 - \langle x, y \rangle} \cdot y.
\end{aligned} \tag{60}$$

   By convention, let $\varphi_x(x) = \infty$, then $\varphi_x \colon \mathbb{S}^d \to H_x \cup \{\infty\}$.

3. The *spherical ray* with root $x$ and direction $y$, denoted by $r_y^x$, is defined as

$$r_y^x = \varphi_x^{-1}\left( \{ t \cdot y \ : \ t > 0 \} \cup \{\infty\} \right). \tag{61}$$

   Each ray $r_y^x$ is isomorphic to $[0, \pi]$ via $d_{\mathbb{S}^d}(x, \cdot)$, so it is parameterized as $(t, r_y^x)$.

4. Spherical trees $\mathcal{T}_{y_1,\ldots,y_k}^x$ in $\mathbb{S}^d$ is the gluing space of $k$ spherical rays $r_{y_i}^x$ at the root $x$. $x$ is the root and $y_1, \ldots, y_k$ are the edges of $\mathcal{T}_{y_1,\ldots,y_k}^x$. It is a measure metric space, endowed with tree metric.

5. The space of spherical trees with $k$ edges in $\mathbb{S}^d$ is denoted by $\mathbb{T}_k^d$, with a probability distribution $\sigma$ on $\mathbb{T}_k^d$, which comes from the tree sampling process.

6. For $\mathcal{T} \in \mathbb{T}_k^d$, *the space of integrable functions on $\mathcal{T}$* is:

$$L^1(\mathcal{T}) = \left\{ f \colon \mathcal{T} \to \mathbb{R} \ : \ \|f\|_{\mathcal{L}} = \sum_{i=1}^k \int_0^\pi |f(t, r_y^x)| \, dt < \infty \right\}. \tag{62}$$

7. A splitting map $\alpha$ is a continuous map from $\mathbb{S}^d \times \mathbb{T}_k^d$ to the $(k-1)$-dimensional standard simplex $\Delta_{k-1}$, i.e. $\alpha \in \mathcal{C}\left( \mathbb{S}^d \times \mathbb{T}_k^d, \Delta_{k-1} \right)$. For $f \in L^1(\mathbb{S}^d)$, we define:

$$\begin{aligned}
\mathcal{R}_{\mathcal{T}}^\alpha f \quad : \quad & \mathcal{T} \quad \longrightarrow \quad \mathbb{R} \tag{63} \\
& (t, r_{y_i}^x) \longmapsto \int_{\mathbb{S}^d} f(y) \cdot \alpha(y, \mathcal{T})_i \cdot \delta(t - \arccos \langle x, y \rangle) \, dy. \tag{64}
\end{aligned}$$

   The function $\mathcal{R}_{\mathcal{T}}^\alpha f$ is in $L^1(\mathcal{T})$.

8. The operator:

$$\begin{aligned}
\mathcal{R}^\alpha \ : \ L^1(\mathbb{S}^d) \longrightarrow & \prod_{\mathcal{T} \in \mathbb{T}_k^d} L^1(\mathcal{T}) \\
f \longmapsto & \ (\mathcal{R}_{\mathcal{T}}^\alpha f)_{\mathcal{T} \in \mathbb{T}_k^d}.
\end{aligned}$$

   is called the *Spherical Radon Transform on Spherical Trees*.

9. When the splitting map $\alpha$ is $O(d+1)$-invariant (this $O(d+1)$-invariance will be described in the next part), the Spherical Radon Transform on Spherical Trees is injective (see (Tran et al., 2025a)).

10. The *Spherical Tree-Sliced Wasserstein Distance* (Tran et al., 2025a) between $\mu, \nu$ in $\mathcal{P}(\mathbb{S}^d)$ is defined by:

$$\text{STSW}(\mu, \nu) = \int_{\mathbb{T}_k^d} W_{d_{\mathcal{T}},1}(\mathcal{R}_{\mathcal{T}}^\alpha f_\mu, \mathcal{R}_{\mathcal{T}}^\alpha f_\nu)\, d\sigma(\mathcal{T}). \tag{65}$$

11. The STSW distance is a metric on $\mathcal{P}(\mathbb{S}^d)$.

12. It is worth noting that, on tree systems, optimal transport problems admits closed-form expression, since it is a metric space with tree metric (see (Le et al., 2019)). Leveraging this closed-form expression and the Monte Carlo method, the distance in Eq. (65) can be efficiently approximated by a closed-form expression.

**The group** $O(d+1)$ **and its actions.** The orthogonal group $O(d+1)$ is the group of linear transformations of $\mathbb{R}^{d+1}$ that preserves the Euclidean norm $\|\cdot\|_2$. The group $O(d+1)$ acts on $\mathbb{S}^d$ naturally as follows: For $x \in \mathbb{S}^d$ and $g = Q \in O(d+1)$, we have:

$$(g, x) \longmapsto gx = Q \cdot x. \tag{66}$$

It naturally induces a group action on the set of all spherical lines in $\mathbb{S}^d$, as well as spherical trees. The tree structure of a spherical tree is preserved under the action of $O(d+1)$ (see (Tran et al., 2025c;a)). In other words, if $\mathcal{T} \in \mathbb{T}$ is a spherical tree, then $g\mathcal{T}$ is also a spherical tree.

**Definition C.1.** A splitting map $\alpha$ in $\mathcal{C}(\mathbb{S}^d \times \mathbb{T}_k^d, \Delta_{k-1})$ is said to be $O(d+1)$-invariant, if we have

$$\alpha(gy, g\mathcal{T}) = \alpha(y, \mathcal{T}) \tag{67}$$

for all $(y, \mathcal{T}) \in \mathbb{S}^d \times \mathbb{T}_k^d$ and $g \in O(d+1)$.

A candidate for $O(d+1)$-invariant splitting maps is presented as follows: Consider the map $\beta \colon \mathbb{S}^d \times \mathbb{T}_k^d \to \mathbb{R}^k$:

$$\beta(y, \mathcal{T}_{y_1,\ldots,y_k}^x)_i = \begin{cases} 0, & \text{if } y = x \text{ or } y = -x, \\ \arccos\left(\dfrac{\langle y, y_i \rangle}{\sqrt{1 - \langle x, y \rangle^2}}\right) \cdot \sqrt{1 - \langle x, y \rangle^2}, & \text{if } y \neq \pm x. \end{cases} \tag{68}$$

The map $\beta$ is continuous and $O(d+1)$-invariant. Take $\alpha \colon \mathbb{S}^d \times \mathbb{T}_k^d \to \Delta_{k-1}$ to be:

$$\alpha(y, \mathcal{T}) = \text{softmax}\left(\{\beta(y, \mathcal{T})_i\}_{i=1,\ldots,k}\right). \tag{69}$$

## C.2. Spatial Spherical Radon Transform on Spherical Trees

Consider a positive integer $d_\theta$, and an injective continuous map $h \colon \mathbb{S}^d \to \mathbb{S}^{d_\theta}$, and a splitting map $\alpha \in \mathcal{C}(\mathbb{R}^{d_\theta} \times \mathbb{L}_k^{d_\theta}, \Delta_{k-1})$ defining the splitting mechanism. Let $\mathcal{T}$ be a spherical tree of $k$ edges in $\mathbb{T}_k^{d_\theta}$. For a function $f \in L^1(\mathbb{S}^d)$, define the function $\mathcal{H}_{\mathcal{T}}^\alpha f \in L^1(\mathcal{T})$ as follows:

$$\mathcal{R}_{\mathcal{T}}^\alpha f \quad : \quad \mathcal{T} \quad \longrightarrow \quad \mathbb{R} \tag{70}$$

$$(t, r_{y_i}^x) \quad \longmapsto \quad \int_{\mathbb{S}^d} f(y) \cdot \alpha(h(y), \mathcal{T})_i \cdot \delta(t - \arccos \langle x, h(y) \rangle)\, dy, \tag{71}$$

The *Spatial Spherical Radon Transform on Spherical Trees* is defined as the operator:

$$\mathcal{H}^\alpha \quad : \quad L^1(\mathbb{R}^d) \quad \longrightarrow \quad \prod_{\mathcal{T} \in \mathbb{T}_k^{d_\theta}} L^1(\mathcal{T}) \tag{72}$$

$$f \quad \longmapsto \quad (\mathcal{H}_{\mathcal{T}}^\alpha f)_{\mathcal{T} \in \mathbb{T}_k^{d_\theta}}.$$

## C.3. Proof for Theorem 4.4

We present the proof for Theorem 4.4 about the injectivity of the Spatial Spherical Radon Transform on Spherical Trees.

*Proof.* Recall the Radon Transform on Spherical Tres $\mathcal{R}^\alpha$ (Tran et al., 2025a) as follows:

$$
\mathcal{R}^\alpha : \; L^1(\mathbb{S}^d) \; \longrightarrow \; \prod_{\mathcal{T} \in \mathbb{T}_k^d} L^1(\mathcal{T})
$$
$$
f \; \longmapsto \; (\mathcal{R}_\mathcal{T}^\alpha f)_{\mathcal{T} \in \mathbb{T}_k^d}, \tag{73}
$$

where

$$
\mathcal{R}_\mathcal{T}^\alpha f : \qquad \mathcal{T} \quad \longrightarrow \quad \mathbb{R}
$$
$$
(t, r_{y_i}^x) \; \longmapsto \; \int_{\mathbb{S}^d} f(y) \cdot \alpha(y, \mathcal{L})_1 \cdot \delta\left(t - \arccos\langle x, y\rangle\right) \, dy, \tag{74}
$$

It is proved in (Tran et al., 2025a) that $\mathcal{R}^\alpha$ is injective for $O(d+1)$-invariant splitting map $\alpha$. We use this result to prove the Spatial Radon Transform on Spherical Trees is injective. First, by the injective continuous map $h : \mathbb{S}^d \to \mathbb{S}^{d_\theta}$, we show that the push-forward of $f \in \mathbb{R}^d$ via $h$, defined as:

$$
h_\sharp f(y) = \begin{cases} f(h^{-1}(y)) & , \; \text{for all } y \in \mathbb{S}^{d_\theta} \; \text{such that } y \in h(\mathbb{S}^d), \\ 0 & , \; \text{for all } y \in \mathbb{S}^{d_\theta} \; \text{such that } y \notin h(\mathbb{S}^d). \end{cases} \tag{75}
$$

has its Spherical Radon Transform on Spherical Trees, i.e. $\{\mathcal{R}_\mathcal{T}^\alpha(h_\sharp f)\}_{\mathcal{T} \in \mathbb{T}_k^{d_\theta}}$, equal to the Spatial Radon Transform on Spherical Trees of $f$, i.e. $\{\mathcal{H}_\mathcal{T}^\alpha f\}_{\mathcal{T} \in \mathbb{T}_k^{d_\theta}}$. In other words, for all $\mathcal{T} \in \mathbb{T}_k^{d_\theta}$, we have:

$$
\mathcal{H}_\mathcal{T}^\alpha f = \mathcal{R}_\mathcal{T}^\alpha(h_\sharp f). \tag{76}
$$

Indeed, we have:

$$
\begin{aligned}
\mathcal{R}_\mathcal{T}^\alpha(h_\sharp f)(t, r_{y_i}^x) &= \int_{\mathbb{S}^{d_\theta}} h_\sharp f(y) \cdot \alpha(y, \mathcal{L})_l \cdot \delta\left(t - \arccos\langle x, y\rangle\right) \, dy \\
&= \int_{h(\mathbb{S}^d)} h_\sharp f(y) \cdot \alpha(y, \mathcal{L})_l \cdot \delta\left(t - \arccos\langle x, y\rangle\right) \, dy \\
&= \int_{\mathbb{S}^d} f(h^{-1}(h(y)) \cdot \alpha(h(y), \mathcal{L})_l \cdot \delta\left(t - \arccos\langle x, h(y)\rangle\right) \, dy \\
&= \int_{\mathbb{S}^d} f(y) \cdot \alpha(h(y), \mathcal{L})_l \cdot \delta\left(t - \arccos\langle x, h(y)\rangle\right) \, dy \\
&= \mathcal{H}_\mathcal{T}^\alpha f(t, r_{y_i}^x).
\end{aligned} \tag{77}
$$

It is clear that $\mathcal{H}^\alpha$ is a linear operator. To prove $\mathcal{H}^\alpha$ is injective, consider $f \in \text{Ker}(\mathcal{H}^\alpha)$. Since $\mathcal{H}_\mathcal{T}^\alpha f = \mathcal{R}_\mathcal{T}^\alpha(h_\sharp f)$, it implies that $h_\sharp f \in \text{Ker}(\mathcal{R}^\alpha)$. Since $\mathcal{R}^\alpha$ is injective, it implies that $h_\sharp f$ is the function 0. By the definition of the push-forward $h_\sharp f$ as in Eq. (75), we conclude that $f$ is the function 0. In conclusion, $\mathcal{H}^\alpha$ is injective. $\qquad\square$

## C.4. Spatial Spherical Tree-Sliced Wasserstein Distance

For two probability measures $\mu$ and $\nu$ with density function $f_\mu$ and $f_\nu$. Given a positive integer $d_\theta$ and a choice of the continuous injective map $h : \mathbb{S}^d \to \mathbb{S}^{d_\theta}$, the *Spatial Spherical Tree-Sliced Wasserstein Distance* between $\mu$ and $\nu$ is defined as the average Wasserstein distance on the tree-metric space $\mathcal{L}$ between the Spatial Spherical Radon Transform on Spherical Trees of $f_\mu$ and $f_\nu$. Following (Tran et al., 2025a), this averaging is taken over the space of trees $\mathbb{T}_k^{d_\theta}$, according to a distribution $\sigma$ on $\mathbb{T}_k^{d_\theta}$ which arises from the tree sampling process.

**Definition C.2.** The *Spatial Spherical Tree-Sliced Wasserstein Distance* between $\mu$ and $\nu$ in $\mathcal{P}(\mathbb{S}^d)$ is defined by:

$$
\text{SpatialSTSW}(\mu, \nu) := \int_{\mathbb{T}_k^{d_\theta}} \text{W}(\mathcal{H}_\mathcal{T}^\alpha f_\mu, \mathcal{H}_\mathcal{T}^\alpha f_\nu) \, d\sigma(\mathcal{T}). \tag{78}
$$

SpatialSTSW is a metric on the space $\mathcal{P}(\mathbb{S}^d)$ of measures on $\mathbb{S}^d$.

**Theorem C.3.** *SpatialSTSW is a metric on the space $\mathcal{P}(\mathbb{S}^d)$.*

*Proof.* We will show that:

$$\text{SpatialSTSW}(\mu, \nu) = \int_{\mathbb{T}_k^{d_\theta}} \text{W}(\mathcal{H}_{\mathcal{T}}^\alpha f_\mu, \mathcal{H}_{\mathcal{T}}^\alpha f_\nu) \, d\sigma(\mathcal{T}), \tag{79}$$

is a metric on $\mathcal{P}(\mathbb{S}^d)$, by verifying its positive definiteness, symmetry and triangle inequality.

**Positive definiteness.** For $\mu, \nu \in \mathcal{P}(\mathbb{S}^d)$, it is clear that

$$\text{SpatialSTSW}(\mu, \mu) = 0, \tag{80}$$

and

$$\text{SpatialSTSW}(\mu, \nu) \geqslant 0. \tag{81}$$

If $\text{SpatialSTSW}(\mu, \nu) = 0$, then $\text{W}(\mathcal{H}_{\mathcal{T}}^\alpha f_\mu, \mathcal{H}_{\mathcal{T}}^\alpha f_\nu) = 0$ for almost every $\mathcal{T} \in \mathbb{T}_k^d$. Since W is a metric on $\mathcal{P}(\mathcal{T})$, we have $\mathcal{H}_{\mathcal{T}}^\alpha f_\mu = \mathcal{H}_{\mathcal{T}}^\alpha f_\nu$ for almost every $\mathcal{L} \in \mathbb{T}$. By the injectivity of the Spatial Spherical Radon Transform on Spherical Trees, it implies that $\mu = \nu$.

**Symmetry.** For $\mu, \nu \in \mathcal{P}(\mathbb{S}^d)$, we have:

$$\begin{aligned}
\text{SpatialSTSW}(\mu, \nu) &= \int_{\mathbb{T}_k^{d_\theta}} \text{W}(\mathcal{H}_{\mathcal{L}}^\alpha f_\mu, \mathcal{H}_{\mathcal{L}}^\alpha f_\nu) \, d\sigma(\mathcal{T}) \\
&= \int_{\mathbb{T}_k^{d_\theta}} \text{W}(\mathcal{H}_{\mathcal{L}}^\alpha f_\nu, \mathcal{H}_{\mathcal{L}}^\alpha f_\mu) \, d\sigma(\mathcal{T}) \\
&= \text{SpatialSTSW}(\nu, \mu)
\end{aligned} \tag{82}$$

So $\text{SpatialSTSW}(\mu, \nu) = \text{SpatialSTSW}(\nu, \mu)$.

**Triangle inequality.** For $\mu_1, \mu_2, \mu_3 \in \mathcal{P}(\mathbb{S}^d)$, we have:

$$\begin{aligned}
&\text{SpatialSTSW}(\mu_1, \mu_2) + \text{SpatialSTSW}(\mu_2, \mu_3) \\
&= \int_{\mathbb{T}_k^{d_\theta}} \text{W}(\mathcal{H}_{\mathcal{T}}^\alpha f_{\mu_1}, \mathcal{H}_{\mathcal{T}}^\alpha f_{\mu_2}) \, d\sigma(\mathcal{T}) + \int_{\mathbb{T}_k^{d_\theta}} \text{W}(\mathcal{H}_{\mathcal{T}}^\alpha f_{\mu_2}, \mathcal{H}_{\mathcal{T}}^\alpha f_{\mu_3}) \, d\sigma(\mathcal{T}) \\
&= \int_{\mathbb{T}_k^{d_\theta}} \left( \text{W}(\mathcal{H}_{\mathcal{T}}^\alpha f_{\mu_1}, \mathcal{H}_{\mathcal{T}}^\alpha f_{\mu_2}) + \text{W}(\mathcal{H}_{\mathcal{T}}^\alpha f_{\mu_1}, \mathcal{H}_{\mathcal{T}}^\alpha f_{\mu_2}) \right) \, d\sigma(\mathcal{T}) \\
&\geqslant \int_{\mathbb{T}_k^{d_\theta}} \text{W}(\mathcal{H}_{\mathcal{T}}^\alpha f_{\mu_1}, \mathcal{H}_{\mathcal{T}}^\alpha f_{\mu_3}) \, d\sigma(\mathcal{T}) \\
&= \text{CircularTSW}(\mu_1, \mu_3).
\end{aligned} \tag{83}$$

The triangle inequality holds for SpatialSTSW.

In conclusion, SpatialSTSW is a metric on the space $\mathcal{P}(\mathbb{S}^d)$. $\square$

**The choice of the injective map $h$.** Note that, the map $h : \mathbb{S}^d \to \mathbb{S}^{d_\theta}$ has to satisfy the injective condition. We construct $h$ as follows. We construct $h$ as follows. First, consider $d_\theta = d + 1$. We define a continuous function: $k(y) = \frac{\pi}{2(1+\epsilon)} \left( \frac{1}{d+1} \sum_{i=0}^{d} y_i + 1 + \epsilon \right)$, which maps $y \in \mathbb{S}^d$ to the range $(0, \pi)$. We set $\epsilon = 10^{-6}$. Using this, we define the mapping $h(y) = (\cos(k(y)), \sin(k(y)) \cdot y)$, which is injective.

## D. Experimental Details

### D.1. Algorithm of proposed Tree-Sliced Distances

We describe the pseudo-codes for $\widehat{\text{CircularTSW}}$, $\widehat{\text{SpatialTSW}}$, $\widehat{\text{SpatialSTSW}}$ in Algorithms 1, 2, 3 respectively.

---

**Algorithm 1** Circular Tree-Sliced Wasserstein distance.

---

**Input:** Probability measures $\mu$ and $\nu$ in $\mathcal{P}(\mathbb{R}^d)$, number of tree systems $L$, number of lines in tree system $k$, space of tree systems $\mathbb{T}$, splitting maps $\alpha$, and parameter $r \in \mathbb{R}_{\geqslant 0}$.
**for** $i = 1$ to $L$ **do**
    Sampling $x \in \mathbb{R}^d$ and $\theta_1, \ldots, \theta_k \overset{i.i.d}{\sim} \mathcal{U}(\mathbb{S}^{d-1})$.
    Contruct tree system $\mathcal{L}_i = \{(x, \theta_1), \ldots, (x, \theta_k)\}$.
    Projecting $\mu$ and $\nu$ onto $\mathcal{L}_i$ to get $\mathcal{CR}^\alpha_{\mathcal{L}_i, r}\mu$ and $\mathcal{CR}^\alpha_{\mathcal{L}_i, r}\nu$.
    Compute $\widehat{\text{CircularTSW}}(\mu, \nu) = (1/L) \cdot \text{W}(\mathcal{CR}^\alpha_{\mathcal{L}_i, r}\mu, \mathcal{CR}^\alpha_{\mathcal{L}_i, r}\nu)$.
**end for**
**Return:** $\widehat{\text{CircularTSW}}(\mu, \nu)$.

---

**Algorithm 2** Spatial Tree-Sliced Wasserstein distance.

---

**Input:** Probability measures $\mu$ and $\nu$ in $\mathcal{P}(\mathbb{R}^{d_\theta})$, number of tree systems $L$, number of lines in tree system $k$, space of tree systems $\mathbb{T}$, splitting maps $\alpha$, and injective continuous map $h : \mathbb{R}^d \to \mathbb{R}^{d_\theta}$.
**for** $i = 1$ to $L$ **do**
    Sampling $x \in \mathbb{R}^d$ and $\theta_1, \ldots, \theta_k \overset{i.i.d}{\sim} \mathcal{U}(\mathbb{S}^{d_\theta - 1})$.
    Contruct tree system $\mathcal{L}_i = \{(x, \theta_1), \ldots, (x, \theta_k)\}$.
    Projecting $\mu$ and $\nu$ onto $\mathcal{T}_i$ to get $\mathcal{H}^\alpha_{\mathcal{L}_i}\mu$ and $\mathcal{H}^\alpha_{\mathcal{L}_i}\nu$.
    Compute $\widehat{\text{SpatialTSW}}(\mu, \nu) = (1/L) \cdot \text{W}(\mathcal{R}^\alpha_{\mathcal{L}_i}\mu, \mathcal{R}^\alpha_{\mathcal{L}_i}\nu)$.
**end for**
**Return:** $\widehat{\text{SpatialTSW}}(\mu, \nu)$.

---

**Algorithm 3** Spatial Spherical Tree-Sliced Wasserstein distance.

---

**Input:** Probability measures $\mu$ and $\nu$ in $\mathcal{P}(\mathbb{S}^d)$, number of tree systems $L$, number of lines in tree system $k$, space of tree systems $\mathbb{T}$, splitting maps $\alpha$, and injective continuous map $h : \mathbb{S}^d \to \mathbb{S}^{d_\theta}$.
**for** $i = 1$ to $L$ **do**
    Sampling $x \in \mathbb{S}^{d_\theta}$ and $y_1, \ldots, y_k \overset{i.i.d}{\sim} \mathcal{U}(\mathbb{S}^{d_\theta - 1})$.
    Contruct tree system $\mathcal{T}_i = \{(x, y_1), \ldots, (x, y_k)\}$.
    Projecting $\mu$ and $\nu$ onto $\mathcal{L}_i$ to get $\mathcal{H}^\alpha_{\mathcal{T}_i}\mu$ and $\mathcal{H}^\alpha_{\mathcal{T}_i}\nu$.
    Compute $\widehat{\text{SpatialSTSW}}(\mu, \nu) = (1/L) \cdot \text{W}(\mathcal{H}^\alpha_{\mathcal{T}_i}\mu, \mathcal{H}^\alpha_{\mathcal{T}_i}\nu)$.
**end for**
**Return:** $\widehat{\text{SpatialSTSW}}(\mu, \nu)$.

---

### D.2. Computational and Memory Complexity

We provide complexity and memory analysis of our proposed distance. Since memory on GPU can be optimized for parallel processing capabilities, we provide the empirical memory usage on GPU, with expectation that our distance would be used in a GPU setting which is standard in machine learning.

**Computation and Memory Complexity.** Assuming $n \geqslant m$, the computational complexity of SpatialTSW and CircularTSW is $O(Lknd_\theta + Lkn \log n)$, while CircularTSW$_{r=0}$ and SpatialSTSW have a more efficient complexity of $O(Lknd_\theta + Ln \log n)$. All distances share an empirical memory cost of $O(Lkn + Lkd_\theta + nd_\theta)$. We analyze the main operations of our proposed distances in Table 6.

Table 6: Computational and Memory Complexity Analysis of proposed Tree-Sliced distances

| Distance | Operation | Description | Computation | Memory |
|---|---|---|---|---|
| SpatialTSW | Mapping | Map points onto new space | $O(nd_\theta)$ | $O(nd_\theta)$ |
| | Projection | Matrix multiplication of points and lines | $O(Lknd_\theta)$ | $O(Lkd_\theta + nd_\theta)$ |
| | Distance-based weight splitting | Distance calculation and softmax | $O(Lknd_\theta)$ | $O(Lkn + Lkd_\theta + nd_\theta)$ |
| | Sorting | Sorting projected coordinates | $O(Lkn \log n)$ | $O(Lkn)$ |
| | **Total** | | $O(Lknd_\theta + Lkn \log n)$ | $O(Lkn + Lkd_\theta + nd_\theta)$ |
| CircularTSW | Circular projection | Subtraction and Norm calculation | $O(Lknd_\theta)$ | $O(Lkd_\theta + nd_\theta)$ |
| | Distance-based weight splitting | Distance calculation and softmax | $O(Lknd_\theta)$ | $O(Lkn + Lkd_\theta + nd_\theta)$ |
| | Sorting | Sorting projected coordinates | $O(Lkn \log n)$ | $O(Lkn)$ |
| | **Total** | | $O(Lknd_\theta + Lkn \log n)$ | $O(Lkn + Lkd_\theta + nd_\theta)$ |
| CircularTSW$_{r=0}$ | Circular projection | Subtraction and Norm calculation | $O(Lnd_\theta)$ | $O(Ld_\theta + nd_\theta)$ |
| | Distance-based weight splitting | Distance calculation and softmax | $O(Lknd_\theta)$ | $O(Lkn + Lkd_\theta + nd_\theta)$ |
| | Sorting | Sorting projected coordinates | $O(Ln \log n)$ | $O(Ln)$ |
| | **Total** | | $O(Lknd_\theta + Ln \log n)$ | $O(Lkn + Lkd_\theta + nd_\theta)$ |
| SpatialSTSW | Mapping | Map points onto new space | $O(nd_\theta)$ | $O(nd_\theta)$ |
| | Projection | Matrix multiplication of points and source | $O(Lnd_\theta)$ | $O(Ld_\theta + nd_\theta)$ |
| | Distance-based weight splitting | Distance calculation and softmax | $O(Lknd_\theta)$ | $O(Lkn + Lkd_\theta + nd_\theta)$ |
| | Sorting | Sorting projected coordinates | $O(Ln \log n)$ | $O(Ln)$ |
| | **Total** | | $O(Lknd_\theta + Ln \log n)$ | $O(Lkn + Lkd_\theta + nd_\theta)$ |

**Projection and Sorting.** In CircularTSW$_{r=0}$ and SpatialSTSW, the projected coordinates within a tree is the same for all lines. As a result, the computational complexity of these two steps in CircularTSW$_{r=0}$ and SpatialSTSW is reduced by a factor of the number of lines in a tree, $k$, compared to SpatialTSW and CircularTSW. This reduction is the primary reason for the computational advantage of CircularTSW$_{r=0}$ and SpatialSTSW.

**Memory Cost of Distance-Based Splitting.** As previously noted in prior work (Tran et al., 2025b), the empirical GPU-optimized memory cost of distance-based splitting is lower than its theoretical estimate due to kernel fusion optimizations. This operation consists of: (1) computing distance vectors from points to lines ($O(Lknd_\theta)$ computation and memory), (2) calculating their norms ($O(Lknd_\theta)$ computation and $O(Lknd_\theta)$ memory), and (3) applying softmax over all lines in each tree ($O(Lkn)$ computation and memory). While the theoretical cost is $O(Lknd_\theta)$ for both computation and memory, we leverage PyTorch's automatic kernel fusion (via 'torch.compile') to merge these steps into a single operation. This enables the distance vectors ($Lkn \times d_\theta$) to be stored in shared GPU memory rather than global memory. As a result, only three matrices need to be stored: a line matrix ($O(Lkd_\theta)$), a support matrix ($O(nd_\theta)$), and a split weight matrix ($O(Lkn)$), reducing overall GPU memory usage to $O(Lkn + Lkd_\theta + nd_\theta)$.

### D.3. Runtime and memory analysis

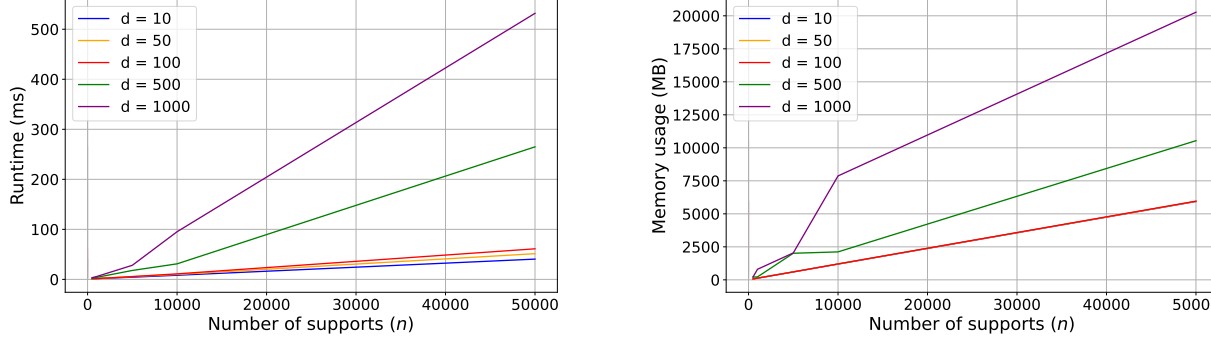

Figure 3: Runtime and memory evolution of SpatialTSW.

In this section, we conduct a runtime and memory analysis of SpatialTSW, CircularTSW, and CircularTSW$_{r=0}$ with

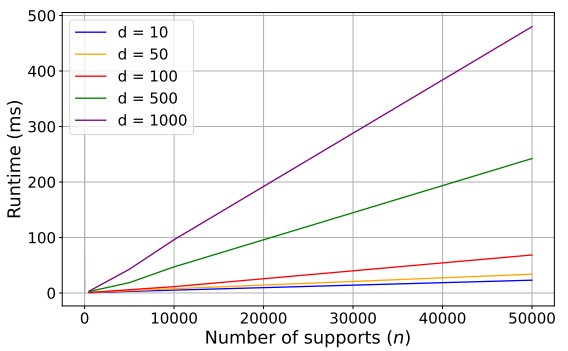 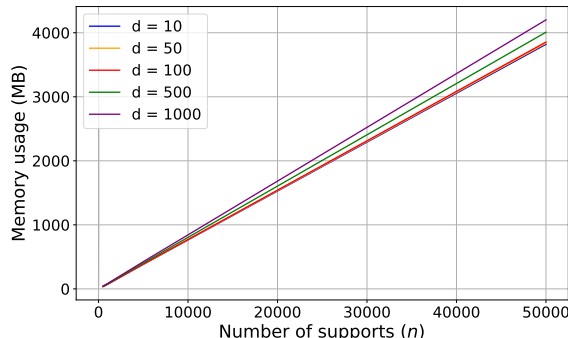

Figure 4: Runtime and memory analysis of CircularTSW.

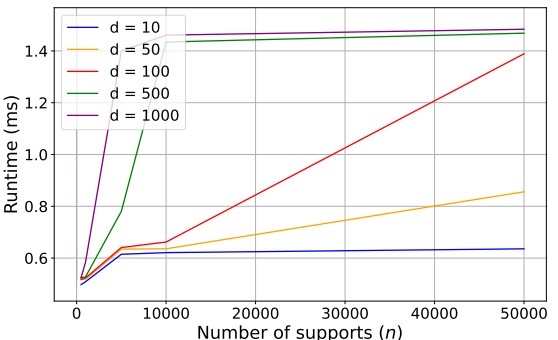 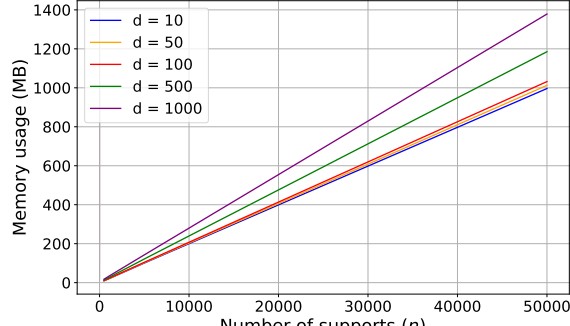

Figure 5: Runtime and memory analysis of CircularTSW$_{r=0}$.

respect to the number of supports and the support's dimension on a single NVIDIA A100 GPU. We fix $L = 2500$ and $k = 4$ (following the practical setting from our diffusion model experiments) for all configurations and vary $N \in \{500, 1000, 5000, 10000, 50000\}$ and $d \in \{10, 50, 100, 500, 1000\}$.

**Runtime scaling.** Figures 3 and 4 demonstrate a linear relationship between the runtime of SpatialTSW and CircularTSW and the number of supports $n$. Regarding scaling with the data dimension, we observe that $d = 10000$ takes approximately twice the runtime of $d = 5000$, suggesting a linear relationship between dimension and computational time. This linear trend aligns with our theoretical complexity analysis in Appendix D.2. It is also worth noting that CircularTSW runs faster than SpatialTSW, as it relies on vector norms instead of vector multiplications. Regarding CircularTSW$_{r=0}$, Figure 5 also demonstrates an almost linear relationship with the number of supports $n$. Interestingly, the runtime of CircularTSW$_{r=0}$ scales very efficiently with $d$. We suspect this is due to the reduced amount of vector normalization required compared to CircularTSW (by a factor of $k$). Ultimately, CircularTSW$_{r=0}$ is significantly faster than other Tree-Sliced methods, by two orders of magnitude when $d$ and $n$ are sufficiently large, which aligns with our theoretical complexity analysis. The significant reduction in computational cost when using the Circular Sliced Wasserstein variants arises from the efficiency of computing $L_2$ norms compared to inner products. This advantage is illustrated in Figure 6.

**Memory scaling.** Figures 3 , 4, and 5 presents the memory consumption analysis of SpatialTSW, CircularTSW, and CircularTSW$_{r=0}$, revealing a linear scaling relationship with $d$ and $n$. This aligns with the theoretical complexity analysis and suggests a predictable scaling behavior. Notably, the peak memory usage of CircularTSW and CircularTSW$_{r=0}$ is significantly lower than that of SpatialTSW.

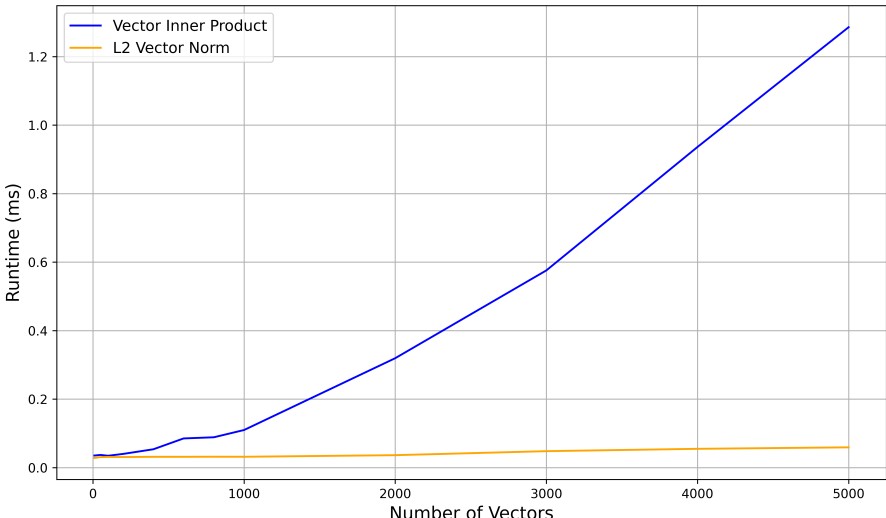

Figure 6: Rebuttal result comparing inner product and L2 norm performance, benchmarked on an A100 GPU with a fixed vector dimension of $d = 3000$. The L2 norm demonstrates significantly faster computation times as the number of vectors increases.

### D.4. Splitting map for CircularTSW

We recall that the splitting map $\alpha$ is defined as:

$$\alpha(y, \mathcal{L})_l = \text{softmax}\Big(\{d(y, \mathcal{L})_i\}_{i=1}^k\Big), \tag{84}$$

where $d(y, \mathcal{L})_i$ represents the distance between $y$ and the $i^{\text{th}}$ line in $\mathcal{L}$.

In the context of the Radon Transform on a system of lines, this involves computing the inner product between the support and the projection directions. However, in the Circular Radon Transform on a system of lines, the projection coordinate calculation involves the Euclidean norm $\|\cdot\|_2$.

Therefore, we define a more computationally efficient distance function as:

$$d(y, \mathcal{L})_i = \|y - x_i - \|y - x_i - r\theta_i\|_2 \, \theta_i\|_2 \,, \tag{85}$$

where $x_i$ is the source, $\theta_i$ is the direction of the $i^{\text{th}}$ line in $\mathcal{L}$, and $r$ is a fixed radius.

Notably, this distance function still results in an $\text{E}(d)$-invariant splitting map.

### D.5. Analysis on number of lines $k$ in CircularTSW$_{r=0}$

In this section, we demonstrate that CircularTSW-DD$_{r=0}$ is effective only in a tree-based setting. To illustrate this, we conduct an experiment using the Denoising Diffusion Generative Adversarial Network (DDGAN), following the setup described in Appendix D.6. Figure 7 presents the FID scores over 300 training epochs for models using CircularTSW$_{r=0}$ with $k = 1$ and $k = 4$. The results clearly show that only the model with $k = 4$ trains stably, with its FID score gradually improving over time. In contrast, for $k = 1$, the FID score suddenly spikes to 500 after epoch 125, indicating that the model collapses and starts generating meaningless images (all black pixels). This confirms that CircularTSW$_{r=0}$ is specifically designed for tree-like structures, relying on a distance-based splitting map to function effectively.

### D.6. Denoising Diffusion Generative Adversarial Network

**Diffusion Models.** Diffusion models (Sohl-Dickstein et al., 2015; Ho et al., 2020) have gained popularity as powerful generative models capable of producing high-quality data. In this experiment, we introduce their mechanisms and demonstrate

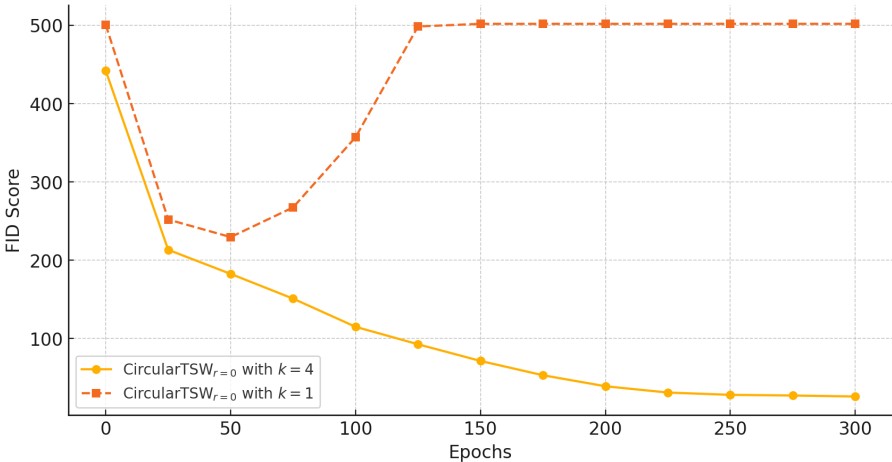

Figure 7: FID scores over the training process for CircularTSW-DD$_{r=0}$ with $k = 1$ and $k = 4$.

the improvements introduced by our approach. The diffusion process begins with an initial sample from the distribution $q(x_0)$ and progressively corrupts it by adding Gaussian noise over $T$ steps. This process is formally defined as:

$$q(x_{1:T}|x_0) = \prod_{t=1}^{T} q(x_t|x_{t-1}),$$

where each transition follows a Gaussian distribution:

$$q(x_t|x_{t-1}) = \mathcal{N}(x_t; \sqrt{1 - \beta_t} x_{t-1}, \beta_t I),$$

with a predefined variance schedule $\beta_t$.

The objective of denoising diffusion models is to learn the reverse diffusion process, which reconstructs the original data from noisy samples. This requires estimating the parameters $\theta$ of the reverse process, formulated as:

$$p_\theta(x_{0:T}) = p(x_T) \prod_{t=1}^{T} p_\theta(x_{t-1}|x_t),$$

where each step follows a Gaussian transition:

$$p_\theta(x_{t-1}|x_t) = \mathcal{N}(x_{t-1}; \mu_\theta(x_t, t), \sigma_t^2 I).$$

Training these models is typically done by maximizing the evidence lower bound (ELBO), which minimizes the Kullback-Leibler (KL) divergence between the true posterior and the model's approximation of the reverse diffusion process. This is expressed as:

$$L = -\sum_{t=1}^{T} \mathbb{E}_{q(x_t)} \left[ \text{KL}(q(x_{t-1}|x_t) || p_\theta(x_{t-1}|x_t)) \right] + C,$$

where $\text{KL}(\cdot || \cdot)$ represents the Kullback-Leibler divergence, and $C$ is a constant term.

**Denoising Diffusion GANs.** While diffusion models generate high-quality and diverse samples, their slow sampling process limits real-world applicability. Denoising Diffusion GANs (DDGANs) (Xiao et al., 2021) address this issue by modeling each denoising step using a multimodal conditional GAN, allowing for larger denoising steps. This significantly reduces the number of steps to just 4, leading to sampling speeds up to 2000 times faster than traditional diffusion models while maintaining competitive sample quality and diversity. The implicit denoising model in DDGANs is formulated as:

$$p_\theta(x_{t-1}|x_t) = \int p_\theta(x_{t-1}|x_t, \epsilon) G_\theta(x_t, \epsilon) d\epsilon, \quad \epsilon \sim \mathcal{N}(0, I).$$

Xiao et al. (2021) optimize the model parameters $\theta$ using adversarial training, with the objective:

$$\min_\phi \sum_{t=1}^{T} \mathbb{E}_{q(x_t)}[D_{adv}(q(x_{t-1}|x_t)||p_\phi(x_{t-1}|x_t))],$$

where $D_{adv}$ represents the adversarial loss. Instead, Nguyen et al. (2024b) replace the adversarial loss with the Augmented Generalized Mini-batch Energy (AGME) distance. For two distributions $\mu$ and $\nu$, given a mini-batch size $n \geqslant 1$, AGME using a Sliced Wasserstein (SW) kernel is defined as:

$$\text{AGME}_b^2(\mu, \nu; g) = \text{GME}_b^2(\tilde{\mu}, \tilde{\nu}),$$

where $\tilde{\mu} = f_\sharp \mu$ and $\tilde{\nu} = f_\sharp \nu$, with $f(x) = (x, g(x))$ for a nonlinear function $g : \mathbb{R}^d \to \mathbb{R}$. The Generalized Mini-batch Energy (GME) distance (Salimans et al., 2018) is defined as:

$$\text{GME}_b^2(\mu, \nu) = 2\mathbb{E}[D(P_X, P_Y)] - \mathbb{E}[D(P_X, P_X')] - \mathbb{E}[D(P_Y, P_Y')],$$

where $X, X' \overset{i.i.d.}{\sim} \mu^{\otimes m}$ and $Y, Y' \overset{i.i.d.}{\sim} \nu^{\otimes m}$, with

$$P_X = \frac{1}{m} \sum_{i=1}^{m} \delta_{x_i}, \quad X = (x_1, \ldots, x_m).$$

Here, $D$ represents any valid distance metric. In our work, we replace $D$ with Tree-Sliced Wasserstein (TSW) variants (our methods) and Sliced Wasserstein (SW) variants.

**Setting.** We adopt the same architecture and hyperparameters as Nguyen et al. (2024b) and Tran et al. (2025b). Our models are trained for 1800 epochs. For Tree-Sliced methods, including our own, we set $L = 2500$ and $k = 4$. For vanilla SW and SW variants, we follow Nguyen et al. (2024b) and use $L = 10000$. The learning rate is also set according to Nguyen et al. (2024b), where $lr_d = 1.25e-4$ and $lr_g = 1.6e-4$. For SpatialTSW, we define $h(y) = y + y^3$, and for CircularTSW, we set $r = 0.01$. The standard deviation in tree sampling follows Tran et al. (2025b) and is set to 0.1. To evaluate runtime, we use a batch size of 64 and measure time on a single NVIDIA A100 GPU.

**Qualitative Results.** Figure 8 presents the qualitative results of SpatialTSW-DD, CircularTSW-DD, and Circular$_{r=0}$TSW-DD.

### D.7. Gradient Flow

**Detailed results on the *25 Gaussians* dataset.** Table 7 presents the detailed results of our proposed methods and baselines. The low standard deviation of SpatialTSW indicates that our method consistently achieves faster convergence compared to other methods.

**Ablation on $h : \mathbb{R}^d \to \mathbb{R}^{d_\theta}$ in SpatialTSW.** We ablate several injective continuous functions $h$ in SpatialTSW on the *25-Gaussian* dataset. The results in Table 8 show that, in general, $h(y) = y + \gamma y^3$ outperforms $h(y) = y + \gamma y^5$, although the latter tends to converge faster during the first 1500 iterations. The best result is achieved with $h(y) = y + 0.5y^3$, yielding a final $W_2$ value of $9.59e-8$.

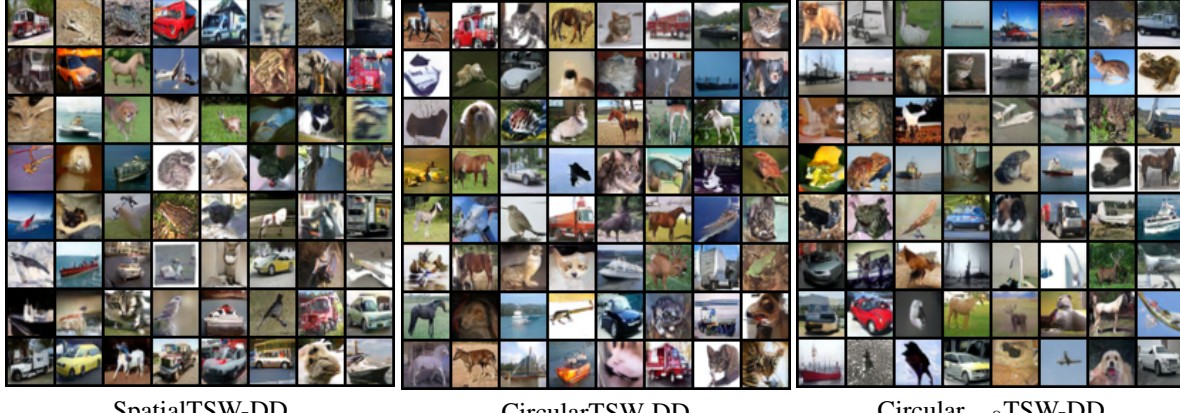

| SpatialTSW-DD | CircularTSW-DD | Circular$_{r=0}$TSW-DD |

Figure 8: Example images generated by our proposed DDGAN. The images correspond to (Left) SpatialTSW-DD, (Middle) CircularTSW-DD, and (Right) Circular$_{r=0}$TSW-DD.

Table 7: Detailed results on the 25 Gaussians dataset. The table reports the average Wasserstein distance between source and target distributions over 5 runs.

| Methods | Iteration | | | | | Time/Iter($s$) |
|---|---|---|---|---|---|---|
| | 500 | 1000 | 1500 | 2000 | 2500 | |
| SW | 4.21e-1 ± 5.39e-3 | 1.54e-1 ± 2.43e-3 | 7.72e-2 ± 3.88e-3 | 4.97e-2 ± 3.30e-3 | 3.59e-2 ± 3.43e-3 | 0.0018 |
| MaxSW | 5.23e-1 ± 8.31e-3 | 2.36e-1 ± 4.63e-3 | 1.23e-1 ± 3.17e-3 | 8.04e-2 ± 3.70e-3 | 6.76e-2 ± 3.07e-3 | 0.1020 |
| SWGG | 6.59e-1 ± 1.93e-2 | 3.62e-1 ± 2.70e-2 | 1.92e-1 ± 1.99e-2 | 9.07e-2 ± 1.31e-2 | 4.42e-2 ± 1.90e-2 | 0.0019 |
| LCVSW | 3.46e-1 ± 4.63e-3 | 6.96e-2 ± 3.11e-3 | 2.26e-2 ± 1.39e-3 | 1.31e-2 ± 2.07e-3 | 9.28e-3 ± 9.25e-4 | 0.0019 |
| TSW-SL | 3.49e-1 ± 4.61e-3 | 8.10e-2 ± 2.34e-3 | 1.06e-2 ± 1.00e-3 | 2.68e-3 ± 3.24e-4 | 3.16e-6 ± 1.99e-6 | 0.0019 |
| Db-TSW | 3.50e-1 ± 5.10e-3 | 8.12e-2 ± 2.34e-3 | 1.09e-2 ± 1.41e-3 | 1.77e-3 ± 6.69e-4 | 1.30e-7 ± 9.28e-9 | 0.0020 |
| Db-TSW$^{\perp}$ | 3.52e-1 ± 5.17e-3 | 7.69e-2 ± 3.37e-3 | 2.73e-2 ± 4.87e-4 | 2.56e-3 ± 7.72e-4 | 2.03e-6 ± 3.70e-6 | 0.0021 |
| SpatialTSW | **3.20e-1 ± 4.73e-3** | **3.44e-2 ± 2.42e-3** | **2.95e-3 ± 1.46e-4** | **3.97e-4 ± 8.13e-5** | **1.17e-7 ± 2.24e-8** | 0.0021 |
| CircularTSW | 4.28e-1 ± 4.33e-3 | 1.20e-1 ± 2.37e-3 | 3.48e-2 ± 5.10e-4 | 1.41e-2 ± 7.50e-4 | 7.86e-3 ± 3.94e-4 | 0.0017 |
| CircularTSW$_{r=0}$ | 4.32e-1 ± 4.01e-3 | 1.22e-1 ± 1.50e-3 | 3.41e-2 ± 2.22e-3 | 1.45e-2 ± 1.03e-3 | 8.94e-3 ± 9.42e-4 | 0.0015 |

**Hyperparameters.** For Tree-Sliced methods, we set $L = 25$ and $k = 4$. For the SW and SW-variant baselines, we use $L = 100$. The global learning rate is set to $0.001$. Each distribution in both datasets is sampled 500 supports.

### D.8. A Guide to Selecting Projection Variants

Our motivation for proposing the non-linear projection framework is inspired by Generalized Sliced-Wasserstein (GSW) (Kolouri et al., 2019), which also includes both Circular and Spatial variants. It is underexplored in prior studies that among the three versions—original SW, SpatialSW, and CircularSW, which variant is most suitable for a given task.

This suggests that among the corresponding TSW variants, such as Db-TSW (Tran et al., 2025b), CircularTSW, and Spatial TSW, there is no guarantee that the versions with non-linear projections will consistently outperform the linear-projection TSW. However, the two new distance variants each offer distinct advantages over standard TSW, as outlined below:

- The definition of SpatialTSW subsumes Db-TSW as a special case when the function is the identity map. This implies that models leveraging SpatialTSW have, in theory, greater representational capacity than those using Db-TSW. A similar relationship holds between the corresponding SW variants.

- The definition of CircularTSW is theoretically non-comparable to Db-TSW due to their fundamentally different constructions. However, CircularTSW$_{r=0}$ offers improved runtime efficiency. This benefit does not hold in the SW context, where CircularSW$_{r=0}$ performs poorly. One reason is that CircularSW$_{r=0}$ defines only a pseudo-metric, while CircularTSW$_{r=0}$ is a true metric.

Table 8: Ablation study on the choice of $h$ in SpatialTSW for Gradient Flow. Results show the average Wasserstein distance between source and target distributions over 5 runs on the 25 Gaussians dataset.

| $h(y)$ | $\gamma$ | Iteration | | | | |
|---|---|---|---|---|---|---|
| | | 500 | 1000 | 1500 | 2000 | 2500 |
| $y + \gamma y^3$ | 0.1 | 3.49e-1 | 7.12e-2 | 7.77e-3 | 6.49e-4 | 1.15e-7 |
| | 0.5 | 3.33e-1 | 4.68e-2 | 3.74e-3 | **1.58e-7** | **9.59e-8** |
| | 1 | 3.20e-1 | 3.44e-2 | 2.95e-3 | 3.97e-4 | 1.17e-7 |
| | 5 | 2.88e-1 | 3.26e-2 | 3.57e-3 | 3.39e-4 | 2.24e-7 |
| | 10 | 2.73e-1 | 3.35e-2 | 5.11e-3 | 1.54e-4 | 5.59e-7 |
| $y + \gamma y^5$ | 0.1 | 3.57e-1 | 7.29e-2 | 7.72e-3 | 1.71e-3 | 1.32e-7 |
| | 0.5 | 3.36e-1 | 4.24e-2 | 4.57e-3 | 1.06e-3 | 1.50e-7 |
| | 1 | 2.99e-1 | 2.37e-2 | 2.95e-3 | 3.50e-5 | 1.63e-7 |
| | 5 | 1.75e-1 | 8.54e-3 | 2.60e-3 | 2.03e-3 | 1.85e-3 |
| | 10 | **1.34e-1** | **6.30e-3** | **3.14e-4** | 6.40e-6 | 1.55e-6 |

Our framework offers greater flexibility by enabling a broader selection of distance functions. However, in Machine Learning, predicting the best variant for a task often requires empirical experimentation. Table 9 shows that both Db-TSW and SpatialTSW perform well, but the non-linearity in SpatialTSW makes it hard to determine in advance which variant is better suited for a given task.

We offer intuition for selecting CircularTSW and CircularTSW$_{r=0}$. Since these distances rely on the $L_2$ norm for the projection step, they are likely to perform well when the $L_2$ norms of the data are diversely distributed. We validate this advantage over Db-TSW and SpatialTSW in the Table 10, where the distribution of $L_2$ norms is uniform. We speculate that this property explains why CircularTSW performs effectively for the Diffusion experiment (Table 1).

To the best of our knowledge, Db-TSW (Tran et al., 2025b) is the only tree-sliced distance effectively suited for large-scale generative tasks involving transport from a training measure to a target measure in Euclidean space. Previously, (Tran et al., 2025c) presents a basic and limited version of (Tran et al., 2025b), primarily emphasizing the constructive aspects of the tree-sliced approach, which serve as foundational groundwork. Meanwhile, (Tran et al., 2025a) explores the method in a spherical setting. Other works on Tree-Sliced Wasserstein (TSW), such as (Le et al., 2019), (Yamada et al., 2022), and others, are mainly designed for classification tasks and are not applicable to generative settings. This limitation arises because these methods rely on a clustering-based framework for computing slices, which is theoretically unsuitable (as the clustering must be recomputed each time the training measure is updated, rendering previous clustering results irrelevant) and empirically inefficient (since clustering is significantly more computationally expensive than linear or non-linear projection methods).

Table 9: Results for Tree-Sliced variants (Linear, Spatial, Circular) in a Gradient Flow task across datasets, showing the average Wasserstein distance between source and target distributions over 5 runs. Each method uses 100 projecting directions, trained for 500 iterations, with the best result reported over $lr \in \{1, 5 \times 10^{-1}, 1 \times 10^{-1}, 1 \times 10^{-2}, 5 \times 10^{-2}, 1 \times 10^{-3}, 3 \times 10^{-3}, 5 \times 10^{-3}\}$. SpatialTSW performs best on Half Moons, Swiss Roll, 25 Gaussians, and 8 Gaussians datasets, while Db-TSW excels on the Circle dataset.

| Methods | Circle | Half Moons | Swiss Roll | 25 Gaussians | 8 Gaussians |
|---|---|---|---|---|---|
| SW | 6.463e-4 ± 3.112e-5 | 1.648e-4 ± 3.754e-5 | 9.795e-4 ± 1.328e-4 | 4.007e-2 ± 1.940e-3 | 3.786e-2 ± 7.090e-3 |
| Db-TSW | **1.331e-6 ± 1.123e-7** | 1.659e-6 ± 1.471e-7 | 1.659e-6 ± 1.471e-7 | 2.103e-3 ± 6.378e-4 | 3.475e-3 ± 1.151e-3 |
| SpatialSW | 2.098e-4 ± 1.919e-5 | 2.146e-4 ± 5.308e-5 | 9.329e-4 ± 1.113e-4 | 5.206e-2 ± 2.456e-3 | 3.593e-2 ± 2.619e-3 |
| SpatialTSW | 1.366e-6 ± 9.439e-8 | **1.267e-6 ± 6.638e-8** | **1.615e-6 ± 1.751e-7** | **1.969e-3 ± 6.314e-4** | **2.652e-3 ± 6.996e-4** |
| CircularSW | 6.044e-4 ± 1.670e-5 | 8.147e-5 ± 4.858e-6 | 7.950e-4 ± 1.065e-4 | 1.150e-1 ± 4.502e-3 | 2.137e-1 ± 1.276e-2 |
| CircularTSW | 1.922e-4 ± 1.493e-5 | 6.982e-5 ± 4.253e-6 | 2.053e-4 ± 3.739e-5 | 1.172e-2 ± 8.564e-4 | 1.307e-2 ± 1.531e-3 |
| CircularTSW$_{r=0}$ | 7.924e-4 ± 5.153e-5 | 9.201e-5 ± 9.562e-6 | 4.030e-4 ± 5.877e-5 | 2.009e-2 ± 1.612e-3 | 3.044e-2 ± 9.105e-4 |

**Hyperparameter $r$.** Selecting the optimal hyperparameter, such as $r$ for CircularTSW, is challenging and often requires empirical tuning. Intuitively, $r$ should be large enough to ensure diverse projections onto the lines but should not exceed the

Table 10: Results on the advantage of the Circular Tree-Sliced variant in a Gradient Flow task when the $L_2$ norm of the data is uniformly distributed. Data is sampled such that the $L_2$ norm follows Uniform$(0, 1)$. The table reports the average Wasserstein distance between source and target distributions over 5 runs. Each method uses 100 projecting directions and is trained for 500 iterations, with the best result reported over $lr \in \{1, 5 \times 10^{-1}, 1 \times 10^{-1}, 1 \times 10^{-2}, 5 \times 10^{-2}, 1 \times 10^{-3}, 3 \times 10^{-3}, 5 \times 10^{-3}\}$. CircularTSW consistently achieves a lower Wasserstein distance, while Linear and Spatial variants struggle as the dimension $d$ increases.

| Methods | $d = 2000$ | $d = 5000$ | $d = 10000$ |
|---|---|---|---|
| SW | $0.535 \pm 0.006$ | $9.795 \pm 0.025$ | $70.06 \pm 0.100$ |
| Db-TSW | $4.871 \pm 0.049$ | $87.49 \pm 0.137$ | $308.97 \pm 0.367$ |
| SpatialSW | $1.510 \pm 0.006$ | $18.66 \pm 0.043$ | $95.55 \pm 0.170$ |
| SpatialTSW | $6.394 \pm 0.066$ | $93.08 \pm 0.131$ | $314.44 \pm 0.269$ |
| CircularSW | $0.357 \pm 0.005$ | $0.404 \pm 0.007$ | $0.428 \pm 0.004$ |
| CircularTSW | $\mathbf{0.304 \pm 0.009}$ | $\mathbf{0.347 \pm 0.010}$ | $\mathbf{0.369 \pm 0.015}$ |
| CircularTSW$_{r=0}$ | $\underline{0.332 \pm 0.010}$ | $0.517 \pm 0.015$ | $0.873 \pm 0.022$ |

data's magnitude. For normalized data, we suggest starting with $r = \frac{1}{\sqrt{d}}$ and tuning from there.

### D.9. Spherical Gradient Flow

**Data.**    Given the probability density function of the von Mises-Fisher distribution $f(x; \mu, \kappa) = C_d(\kappa) \exp(\kappa \mu^T x)$, where $\mu \in \mathbb{S}^d$ is mean direction and $\kappa > 0$ is concentration parameter and the normalization constant $C_d(\kappa) = \dfrac{\kappa^{d/2-1}}{(2\pi)^{p/2} I_{p/2-1}(\kappa)}$, we use 12 vMFs as the target distribution with $\kappa = 50$ and mean directions as follows:

$$
\begin{aligned}
\mu_1 &= (-1, \phi, 0), & \mu_2 &= (1, \phi, 0), & \mu_3 &= (-1, -\phi, 0), & \mu_4 &= (1, -\phi, 0) \\
\mu_5 &= (0, -1, \phi), & \mu_6 &= (0, 1, \phi), & \mu_6 &= (0, -1, -\phi), & \mu_8 &= (0, 1, -\phi) \\
\mu_9 &= (\phi, 0, -1), & \mu_{10} &= (\phi, 0, 1), & \mu_{11} &= (-\phi, 0, -1), & \mu_{12} &= (-\phi, 0, 1),
\end{aligned}
$$

where $\phi = \dfrac{1 + \sqrt{5}}{2}$.

Similar to Tran et al. (2024a; 2025a), we pick 200 samples from each vMF.

**Setting.**    We set $L = 200$ trees and $k = 5$ lines for STSW and SpatialSTSW, and $L = 1000$ projections for other sliced methods. ARI-S3W (30) employs 30 rotations with a pool size of 1000, whereas RI-S3W (1) and RI-S3W (5) use 1 and 5 rotations, respectively. Training is conducted using Adam (Kinga et al., 2015) optimizer with learning rate $lr = 0.01$, following update rules (Bonet et al., 2022):

$$
\begin{cases}
x^{(k+1)} = x^{(k)} - \gamma \nabla_{x^{(k)}} \text{SpatialSTSW}(\hat{\mu}_k, \nu), \\
x^{(k+1)} = \frac{x^{(k+1)}}{\|x^{(k+1)}\|_2}.
\end{cases}
$$

### D.10. Self-Supervised Learning

**Encoder.**    Following the approach outlined in Bonet et al. (2022); Tran et al. (2024a; 2025a), we use ResNet18 (He et al., 2016) as the encoder. We train it on CIFAR-10 data for 200 epochs with a batch size of 512 and the SGD optimizer with initial $lr = 0.05$, momentum 0.9, and weight decay $10^{-3}$. To generate positive pairs, we employ commonly used augmentation techniques, consistent with previous studies (Wang & Isola, 2020; Bonet et al., 2022; Tran et al., 2024a; 2025a). These transformations include resizing, cropping, horizontal flipping, color jittering, and random grayscale conversion.

For STSW and SpatialSTSW, we configure $L = 200$ trees and $k = 20$. For other distances, we use $L = 200$ projections and $N_R = 5$ with a pool size of 100 for RI-S3W and ARI-S3W. The regularization coefficients are chosen as follows: $\lambda = 10$ for STSW and SpatialSTSW, $\lambda = 1$ for SW, $\lambda = 20$ for SSW, and $\lambda = 0.5$ for S3W variants.

**Linear Classifier.** To evaluate the quality of the feature representations learned by the pre-trained encoder, a linear classifier is trained on top of these features. Following the approach of Bonet et al. (2022), the classifier is trained for 100 epochs using the Adam (Kinga et al., 2015) optimizer. We set the initial learning rate to $10^{-3}$ together with a weight decay of $0.2$ at epochs 60 and 80.

### D.11. Sliced-Wasserstein Autoencoder (SWAE)

**Setup.** For our model training, we use Adam (Kinga et al., 2015) optimize, setting learning rate to $lr = 10^{-3}$. The model undergoes training over 100 epochs with a batch size of 500, where the binary cross-entropy (BCE) loss function serves as the reconstruction loss. For STSW and SpatialSTSW, we fix $L = 200$ trees and $k = 10$ lines. Other sliced methods use $L = 100$ projections. RI-S3W ad ARI-S3W use $N_R = 5$ rotation and pool size of 100. We use prior 10 vMFs, while setting the regularization parameter $\lambda = 1$ for STSW, SpatialSTSW, $\lambda = 10$ for SSW, and $\lambda = 10^{-3}$ for SW and S3W variants.

**CIFAR-10 Model Architecture.** (Tran et al., 2024a; 2025a)

Encoder:

$$
\begin{aligned}
x \in \mathbb{R}^{3 \times 32 \times 32} &\to \text{Conv2d}_{32} \to \text{ReLU} \to \text{Conv2d}_{32} \to \text{ReLU} \\
&\to \text{Conv2d}_{64} \to \text{ReLU} \to \text{Conv2d}_{64} \to \text{ReLU} \\
&\to \text{Conv2d}_{128} \to \text{ReLU} \to \text{Conv2d}_{128} \to \text{Flatten} \\
&\to \text{FC}_{512} \to \text{ReLU} \to \text{FC}_3 \\
&\to \ell^2 \text{ normalization} \to z \in \mathbb{S}^2
\end{aligned}
$$

Decoder:

$$
\begin{aligned}
z \in \mathbb{S}^2 &\to \text{FC}_{512} \to \text{FC}_{2048} \to \text{ReLU} \\
&\to \text{Reshape}(128 \times 4 \times 4) \to \text{Conv2dT}_{128} \to \text{ReLU} \\
&\to \text{Conv2dT}_{64} \to \text{ReLU} \to \text{Conv2dT}_{64} \to \text{ReLU} \\
&\to \text{Conv2dT}_{32} \to \text{ReLU} \to \text{Conv2dT}_{32} \to \text{ReLU} \\
&\to \text{Conv2dT}_3 \to \text{Sigmoid}
\end{aligned}
$$

### D.12. Hardware settings

The gradient flow experiments were conducted on a single NVIDIA A100 GPU, with each experiment taking approximately 0.5 hours. The denoising diffusion experiments were executed in parallel on two NVIDIA A100 GPUs, with each run lasting around 50 hours. All spherical experiments were conducted on a single NVIDIA A100 GPU.

