# OpenReview forum: "Tree-Sliced Wasserstein Distance with Nonlinear Projection"
_ICML.cc/2025/Conference — ICML 2025 poster_

### Official Review · Reviewer_kfHL · 2025-02-23

**Overall Recommendation:** 4

**Summary:**

The authors introduce the following:
1. Generalized Radon transforms in the system of lines.
- These transforms extend the concept of the Radon transform by incorporating systems of lines and allowing for nonlinear projections, which improve the flexibility and applicability of the SW distance.
2. Generalized Tree-sliced Wasserstein distance.
- The authors develop two specific variants of TSW—the Circular Tree-Sliced Wasserstein distance and the Spatial Tree-Sliced Wasserstein distance, which offer more efficient metrics for measures on both Euclidean spaces and spheres.

3. Applied generalized tree-sliced Wasserstein distance in generative model and gradient flow problems.
In addition, the new distance is applied for spherical data (Gradient flow and self-supervised learning).

**Claims And Evidence:**

Yes.

**Essential References Not Discussed:**

Bonet, C., Berg, P., Courty, N., Septier, F., & Drumetz, L. (2022).
Spherical Sliced-Wasserstein.
arXiv:2206.08780 | PDF
DOI: 10.48550/arXiv.2206.08780

Leluc, R., Dieuleveut, A., Portier, F., & Segers, J. (2024).
Sliced-Wasserstein Estimation with Spherical Harmonics as Control Variates.
arXiv:2402.01493 | PDF
DOI: 10.48550/arXiv.2402.01493

**Experimental Designs Or Analyses:**

Yes. I've checked the experiment design.

**Methods And Evaluation Criteria:**

Yes.

**Other Comments Or Suggestions:**

N/A

**Other Strengths And Weaknesses:**

### Weaknesses:

1. It seems that any continuous injective function \( h \) can be used to define the generalized Radon transform. However, the criteria for selecting a suitable \( h \) for different datasets or tasks remain unclear.
   In Section 5.2, the authors explain that \( h(x) \) can be determined by a neural network, but the structure and size of the neural network are not specified.

2. When \( h \) is implemented as a neural network, the cost of optimizing its parameters is not included in the computational complexity analysis in Section 5.3. This suggests that the actual complexity should be higher than the proposed value.
   Additionally, in Figure 2, the computational cost of circular TSW appears to increase significantly when \( n \geq 400 \), but no explanation for this behavior is provided.

**Questions For Authors:**

1. The Circular Radon Transform (Equation (9)) is defined in \(\mathbb{R}^d\) rather than in a circular space. Could you clarify why it is referred to as the Circular Radon Transform?
   Additionally, what is the reasoning behind the significant reduction in computational cost when using the Circular Radon Transformed Sliced-Wasserstein distance?

**Relation To Broader Scientific Literature:**

**References:**

 Kolouri, S., Park, J., & Rohde, N. (2019). Generalized Sliced Wasserstein Distances. arXiv preprint [arXiv:1906.06962](https://arxiv.org/abs/1906.06962).

Bonet, C., Berg, P., Courty, N., Septier, F., & Drumetz, L. (2022).
Spherical Sliced-Wasserstein.
arXiv:2206.08780 | PDF
DOI: 10.48550/arXiv.2206.08780

Tran, H., Bai, Y., Kothapalli, A., Shahbazi, A., & Liu, X. (2024).
Stereographic Spherical Sliced Wasserstein Distances.
arXiv:2402.02345 | PDF
DOI: 10.48550/arXiv.2402.02345

Leluc, R., Dieuleveut, A., Portier, F., & Segers, J. (2024).
Sliced-Wasserstein Estimation with Spherical Harmonics as Control Variates.
arXiv:2402.01493 | PDF
DOI: 10.48550/arXiv.2402.01493

Relation: Previous works have explored the combination of the generalized Radon transform/spherical Radon transform with the Wasserstein distance. This paper examines the application of the generalized Radon transform/spherical Radon transform to a system and analyzes the corresponding Wasserstein distance.

**Theoretical Claims:**

New concepts:
- Generalized  Radon transform in line system (Section 3.1, Eq(9)) and related injective property (Theorem 4.2-4.4)
- non-linear tree sliced Wasserstein distance (Eq (17), (18)) and metric property (Theorem 5.3)
- Additional RSpatial Spherical Radon Transform on Spherical Trees

---

> ### Author Rebuttal · Authors · 2025-03-31
>
> We direct the Reviewer to Tables R1-2 and Figure R1, available at https://sites.google.com/view/nonlinear-tsw-4.
>
> **Q1. It seems that any continuous injective function $h$ can be used to define the generalized Radon transform. However, the criteria for selecting a suitable $h$ for different datasets or tasks remain unclear.**
>
> **Answer Q1.** We kindly refer the Reviewer to our response to **Q1+W1** in Reviewer m8Ge's review.
>
> **Q2. In Section 5.2, the authors explain that $h(x)$ can be determined by a neural network, but the structure and size of the neural network are not specified.**
>
> **When $h$ is implemented as a neural network, the cost of optimizing its parameters is not included in the computational complexity analysis in Section 5.3. This suggests that the actual complexity should be higher than the proposed value.**
>
> **Answer Q2.** A potential neural network for implementing $h(x)$ can be inspired by [6], where the neural network is a single feedforward layer that maps $\mathbb{R}^d \to \mathbb{R}^d$. Such a layer would have $d \times d$ weights and $d$ biases, introducing $O(d^2)$ parameters. If $h$ is implemented as a neural network, the cost of optimizing its parameters would scale the overall complexity linearly by the number of optimization steps. Incorporating learnable parameters into $h$ could enhance performance by better adapting to the data distribution, but it substantially increases the computational cost.
>
> In our work, efficiency is a major concern, so we opted for a simple mapping to avoid this overhead. Specifically, for SpatialTSW, we use the mapping $h(x) = (f_1(x), \ldots, f_d(x))$, where $f_i(x) = x_i + x_i^3$. This mapping costs $O(n \cdot d_{\theta})$, which is trivial compared to other steps like projection or sorting.
>
> **Q3. What is the reasoning behind the significant reduction in computational cost when using the Circular Radon Transformed Sliced-Wasserstein distance?**
>
> **Additionally, in Figure 2, the computational cost of circular TSW appears to increase significantly when $n \geq 400$, but no explanation for this behavior is provided.**
>
> **Answer Q3.** The significant reduction in computational cost when using the Circular Sliced Wasserstein variants arises from the efficiency of computing $L_2$ norms compared to inner products. This advantage is illustrated in Figure R1.
>
> As shown in Figure 2, the computational cost of $CircularTSW_{r=0}$ increases when $n \geq 400$. This behavior is attributed to the cost of computing the splitting map $\alpha$, which remains $\mathcal{O}(L k n d_{\theta})$ — the same complexity as other tree-sliced variants. While $CircularTSW_{r=0}$ is theoretically and empirically more efficient in the projection and sorting steps, the cost of computing $\alpha$ becomes dominant as the number of support points $n$ grows.
>
> In practice, $CircularTSW_{r=0}$ is slower than standard SW when the number of supports increases significantly since it involves the additional computation of the splitting map $\alpha$, whereas SW does not. However, in a large-scale experiment such as a diffusion model, $CircularTSW_{r=0}$ increases training time by only $12$\% compared to SW while substantially improving the FID from $3.64$ to $2.48$, highlighting a favorable practical trade-off. Additionally, $CircularTSW_{r=0}$ outperforms the best existing tree-sliced method, Db-TSW, in training time and FID.
>
> **Q4. The Circular Radon Transform (Equation (9)) is defined in $\mathbb{R}^d$ rather than in a circular space. Could you clarify why it is referred to as the Circular Radon Transform?**
>
> **Answer Q4.**  The term circular refers to the circular defining function used in the Radon Transform, rather than the domain of the space itself. This naming convention is consistent with several prior works (see [4], [5]).
>
> In contrast, for distances involving measures defined on the sphere, the term spherical is typically used (see [1], [2], [3]).
>
> ---
> We thank the Reviewer for the constructive feedback, as well as for pointing out typos and missing references, which we will address.  If the Reviewer finds our clarifications satisfactory, we kindly ask you to consider raising the score. We would be happy to address any further concerns during the next stage of the discussion.
>
> ---
> *References.*
>
> [1] Hoang Tran et al., Spherical Tree-Sliced Wasserstein Distance, ICLR 2025.
>
> [2] Clément Bonet et al., Spherical Sliced-Wasserstein, ICLR 2023.
>
> [3] Huy Tran et al., Stereographic Spherical Sliced Wasserstein Distances, ICML 2024.
>
> [4] Soheil Kolouri et al., Generalized Sliced Wasserstein Distances, NeurIPS 2019.
>
> [5] Gaik Ambartsoumian et al., On the injectivity of the circular Radon transform, Inverse Problems 21 (2005).
>
> [6] Xiongjie Chen et al., Augmented Sliced Wasserstein Distances, ICLR 2022.

---

### Official Review · Reviewer_jNth · 2025-03-01

**Overall Recommendation:** 3

**Summary:**

This work proposes to extend the Tree Sliced-Wasserstein distances, defined using linear projections on system of lines, by using nonlinear projections instead. The authors study the use of two different non linear projections: circular projections and spatial projections. They also propose to use a spatial projection on the sphere. Then, they introduce the associated Radon transform on system of lines, show that they are injective, define the resulting Tree-Sliced Wasserstein distances and show that these are well distances. Finally, they benchmark these distances on several applications such as generative modeling with Denoising Diffusion GANs and gradient flows for the Euclidean non linear projections, and gradient flows, self-supervised learning and Sliced-Wasserstein autoencoders for the spherical version.

## Update after rebuttal

I maintain my positive score.

**Claims And Evidence:**

The claims made are well supported. The constructions proposed are well justified, and the proofs of the claims such as distance properties are provided.

**Essential References Not Discussed:**

All the essential references seem to be discussed.

**Experimental Designs Or Analyses:**

The experimental designs are good.

**Methods And Evaluation Criteria:**

The evaluation criteria to compare the new distances with previously proposed distances make sense.

**Other Comments Or Suggestions:**

I would suggest to try to improve the introduction of the background on the Tree SW distances, and to add Figures to better understand how the projections work. For instance, in Section 3.2, the Radon transforms on System of Lines are introduced, but all notations do not seem to be introduced (e.g. it is not clear directly what is $\mathbb{L}_k^d$).

In Figure 1, it is not directly clear what is projected, what is $x_i$... It would be better to add a legend on the Figure with labels on the points projected and the center $x_i$. Also, it would maybe help to see a comparison with $r>0$.

In Section 5, it is stated that explanations are provided for the choice of the projections, and why they lead to more efficient metrics, but it is not very clear where it is stated after.

In Figure 2, the result are given for $n$ up to $n=500$, which is rather small for sliced settings.

In Table 2 and 3, this is basically the same experiment, but the results are not provided in the same way ($W_2$ versus $\log_2 W_2$).

Typos:
- Line 204, 1st column: "$R^n$"
- Line 215, 2nd column: "We also examine different choices of functions that define the nonlinear projections explain why certain choices lead to more efficient metrics."
- Line 293, 1st column: "$f_i(x)=x_i+x_i^3$"
- Line 383, 2nd column: "Inspised"

**Other Strengths And Weaknesses:**

This paper introduces on one hand non linear projections to extend the Euclidean and Spherical Tree Sliced-Wasserstein distances, which are natural things to study. This paper is doing it well as it provides justifications to all the choices. However, the background is hard to follow, as it relies a lot on the previous papers [1,2,3], with a brief description in Appendix. This is the main limit that I found with the current state of this work.

**Strengths**:
- Provide new (Spherical) Tree SW distances using non linear projections, which are well distances. In particular, the Spatial Spherical TSW distance use a new projection.
- Several experiments showing benefits compared to other distances

**Weaknesses**:
- The paper feels a bit incremental, but there are lots of results, which compensate.
- The background is not very clear, and lot of important things to understand the paper are in Appendix. Also, more figures could help understanding the constructions.
- It is not really stated when one would prefer one type of non linear projection compared to another.

[1] Tran, V.-H., Pham, T., Tran, T., Le, T., and Nguyen, T. M. Tree-sliced wasserstein distance on a system of lines. arXiv preprint arXiv:2406.13725, 2024.

[2] Tran, H. V., Nguyen-Nhat, M.-K., Pham, H. T., Chu, T., Le, T., and Nguyen, T. M. Distance-based tree-sliced wasserstein distance. In The Thirteenth International Conference on Learning Representations, 2025.

[3] Tran, H. V., Chu, T., Nguyen-Nhat, M.-K., Pham, H. T., Le, T., and Nguyen, T. M. Spherical tree-sliced wasserstein distance. In The Thirteenth International Conference on Learning Representations, 2025.

**Questions For Authors:**

1. You are proposing new choices for the polynomials $h$: how does it compare to the construction proposed in [1]?
2. How the trees are constructed? I am not sure it is precised (but I may have missed it)

[1] Kolouri, S., Nadjahi, K., Simsekli, U., Badeau, R., and Rohde, G. Generalized sliced wasserstein distances. Advances in neural information processing systems, 32, 2019.

**Relation To Broader Scientific Literature:**

The key contribution of this paper is to extend the Tree Sliced-Wasserstein distance proposed in [1, 2] using non linear projections. Non linear projections were first used in [1] for the Sliced-Wasserstein distance. In particular, they leverage results from [2] to show that the proposed constructions are well distances.

The second key contribution is to extend the spherical Tree Sliced-Wasserstein distance proposed in [4] with non linear projections.


[1] Tran, V.-H., Pham, T., Tran, T., Le, T., and Nguyen, T. M. Tree-sliced wasserstein distance on a system of lines. arXiv preprint arXiv:2406.13725, 2024.

[2] Tran, H. V., Nguyen-Nhat, M.-K., Pham, H. T., Chu, T., Le, T., and Nguyen, T. M. Distance-based tree-sliced wasserstein distance. In The Thirteenth International Conference on Learning Representations, 2025.

[3] Kolouri, S., Nadjahi, K., Simsekli, U., Badeau, R., and Rohde, G. Generalized sliced wasserstein distances. Advances in neural information processing systems, 32, 2019.

[4] Tran, H. V., Chu, T., Nguyen-Nhat, M.-K., Pham, H. T., Le, T., and Nguyen, T. M. Spherical tree-sliced wasserstein distance. In The Thirteenth International Conference on Learning Representations, 2025.

**Theoretical Claims:**

The proofs seem to be correct.

---

> ### Author Rebuttal · Authors · 2025-03-31
>
> We direct the Reviewer to Table R1-2 at https://sites.google.com/view/nonlinear-tsw.
>
> **Q1. The paper feels a bit incremental, but there are lots of results, which compensate.**
>
> **It is not really stated when one would prefer one type of non linear projection compared to another.**
>
> **Answer Q1.**  We kindly refer the Reviewer to our response to **Q1+W1** in Reviewer m8Ge's review.
>
> **Q2. You are proposing new choices for the polynomials $h$: how does it compare to the construction proposed in [1]?**
>
> **Answer Q2.** In [1], the author proposed using homogeneous polynomials of odd degree for the mapping $h: \mathbb{R}^d \to \mathbb{R}^{d_\theta}$, where the output dimension $d_\theta$ grows exponentially with $d$. For large $d$, such as in our Diffusion Experiment where $d \approx 3000$, this results in $d_\theta \approx 4.5 \times 10^9$ (see line 263-270), making the approach computationally impossible. In contrast, our proposed mapping, defined as $h(x) = (f_1(x), \ldots, f_d(x))$ with $f_i(x) = x_i + x_i^3$, maintains the same output dimension as the input ($d_\theta = d$) and introduces a trivial computational cost while still ensuring a non-linear projection.
>
> **Q3. How the trees are constructed? I am not sure it is precised (but I may have missed it)**
>
> **Answer Q3.** The tree structure used in our paper is the concurrent-lines structure introduced in [2]. We will make this explicit in the revised version of the paper.
>
> **Q4. The background is not very clear, and lot of important things to understand the paper are in Appendix. Also, more figures could help understanding the constructions.**
>
> **Answer Q4.**
>
> We acknowledge that the background for the tree-sliced approach can be dense, as it builds upon the foundations laid in [2] and [4]. We thank the Reviewer for pointing this out and will revise the paper to include additional figures to improve readability.
>
> **Q5. In Figure 2, the result are given for $n$ up to $n=500$, which is rather small for sliced settings.**
>
> In Figure 2, we use $n$ up to $500$ because Diffusion Model experiments typically train with batch sizes $n < 500$. Additionally, we set $d = 3000$, $L = 2500$, and $k = 4$, aligning with the practical settings of the Diffusion Model experiment. We also provide runtime and memory analysis for our Tree-Sliced Wasserstein variants in Appendix D.3, with $n$ up to $50000$.
>
> **Q6. In Table 2 and 3, this is basically the same experiment, but the results are not provided in the same way ($W_2 \text{ versus } \log_2 W_2$).**
>
> Table 2 is about Gradient Flow on Euclidean space $\mathbb{R}^d$, while Table 3 is about Gradient Flow on the sphere $\mathbb{S}^d$. Therefore, the choice of experimental setting and baselines are different. Note that Table 2 follows the setting of [2], while Table 3 follows the setting of [4].
>
> ---
> We thank the Reviewer for the constructive feedback, as well as for pointing out typos and missing references. We will address them accordingly. If the Reviewer finds our clarifications satisfactory, we kindly ask you to consider raising the score. We would be happy to address any further concerns during the next stage of the discussion.
>
> ---
> *References.*
>
> [1] Soheil Kolouri et al., Generalized Sliced Wasserstein Distances. NeurIPS 2019.
>
> [2] Hoang Tran et al., Distance-Based Tree-Sliced Wasserstein Distance, ICLR 2025.
>
> [3] Hoang Tran et al., Tree-Sliced Wasserstein Distance on a System of Lines.
>
> [4] Hoang Tran et al., Spherical Tree-Sliced Wasserstein Distance, ICLR 2025.
>
> [5] Tam Le et al., Tree-Sliced Variants of Wasserstein Distances, NeurIPS 2019.
>
> [6] Tam Le et al., Sobolev Transport: A Scalable Metric for Probability Measures with Graph Metrics, AISTATS 2022.
>
> [7] Makoto Yamada et al., Approximating 1-Wasserstein Distance with Trees, TMLR 2022.

---

### Official Review · Reviewer_zCq9 · 2025-03-12

**Overall Recommendation:** 2

**Summary:**

This paper extends the Tree-Sliced Wasserstein (TSW) distance, an alternative to the Sliced Wasserstein (SW) distance that leverages tree-based metric spaces, by allowing the use of nonlinear projections. More precisely, the authors explore generalized Radon transforms (previously used in existing SW variants, such as Generalized SW by Kolouri et al. (2019) or Augmented SW by Chen et al. (2022)), and analyze how these can be integrated into TSW while preserving injectivity and invariance properties. This analysis leads to the definitions of two instances of TSW, called "Circular Tree-Sliced Wasserstein Distance" and "Spatial Tree-Sliced Wasserstein Distance". Finally, they apply these new metrics to generative models, gradient flows and self-supervised learning, to compare the result quality and computational efficiency against SW and variants.

**Claims And Evidence:**

The theoretical and methodological contributions of this paper seem sound to me, as they naturally extend related work by combining results from SW based on nonlinear projections and tree-based SW. That said, some imprecise points remain and could be addressed (see section 'Theoretical Claims').

My biggest concern is with the empirical analysis and conclusions. In my opinion, the authors make overclaims that are not properly justified by solid and convincing evidence: instead of providing a more nuanced description of the obtained results, they draw conclusions that seem overstated to me. See my detailed comments in "Methods and Evaluation Criteria" and "Experimental Designs or Analyses".

**Essential References Not Discussed:**

The related works is adequately discussed. Some references that are relevant and seem to be missing are:
- "Parallelly Sliced Optimal Transport on Spheres and on the Rotation Group", M. Quellmalz, L. Buecher, G. Steidl (2024)
- "Sliced Optimal Transport on the Sphere", M. Quellmalz, R. Beinert, G. Steidl (2023)

**Experimental Designs Or Analyses:**

The authors claim that the proposed tree-sliced-Wasserstein distances "consistently outperform state-of-the-art Sliced Wasserstein and Tree-Sliced Wasserstein methods across various tasks" and support this point with a series of experiments (Section 6 and supplementary doc). However, I find the empirical results not convincing enough:

- Across all experiments, the improvement in precision appears marginal (see Tables 1-5), and there is no uncertainty quantification. Additionally, the only qualitative results available (Figure 7 in the supplementary document) do not clearly demonstrate the advantages of the proposed tree-sliced metrics in terms of image generation quality.

- The authors state that their approach "maintains comparable or improved runtime efficiency" and specifically, they report that "CircularTSW-DD and CircularTSW reduce training time relative to Db-TSW-DD⊥ by 10% and 19%, respectively" and "CircularTSW and CircularTSWr=0 are approximately 5% and 16% faster than vanilla SW, respectively". While these claims are accurate, the results in Tables 1 and 2 reveal the reduction in computation time is negligible: CircularTSW and CircularTSWr=0 take 0.0017s and 0.0015s per iteration, respectively, compared to 0.0018s for SW. Furthermore, the results lack consistency across experiments: in Table 1 and Table 5, vanilla SW is the first or second fastest method.

**Methods And Evaluation Criteria:**

The experiments in this paper involve incorporating different metrics into existing black-box machine learning pipelines, whose behavior is hard to interpret. As a result, directly comparing the performance of different variants is challenging, and the analysis relies solely on quantitative scores such as FID, which I find insufficient to illustrate the authors' strong conclusions.

I think the paper would be significantly more convincing if the authors identified a simpler and more interpretable setting where their proposed non-linear tree SW metrics demonstrably capture relevant data features more efficiently. Such a well-chosen controlled experiment could provide more insights into the advantages of their approach, that nicely complement the current empirical analysis.

**Other Comments Or Suggestions:**

- l.98: the definition of the set $L^1(\cdot)$ is missing
- l.107: missing definition for $\mathcal{U}(\mathbb{S}^{d-1})$
- Missing definition for $\theta_\sharp$ (Section 2)
- l.102 : $\mathbb{R}_{\geq 0}$ can be written $\mathbb{R}_+$
- Section 3.2: all the notations given in Section A to define tree-sliced SW should be recalled in that section.
- Equation (14): the operation $gy,g\mathcal{L}$ is unclear in the main text: equation (66) should appear earlier.
- l.709: "Sysmtes"
- Section 6.2: "Inspised"

**Other Strengths And Weaknesses:**

I find the paper lacks clarity and rigor:
- Several key definitions are missing from the main text, making it difficult to read and understand (see "Other comments or suggestions" for details).
- Some strong claims about important aspects of the method lack proper and precise justification. For instance, the notions of well-definedness and injectivity (l.134) are not properly defined, and it is unclear why "injectivity is typically required for Radon Transform variants". Remark 3.2 presents intriguing, non-trivial information, without supporting results or references: why does $\alpha$ induce a tradeoff between effectiveness and theoretical guarantees, , and what kind of guarantees are being referred to? In what sense "the distances derived from RTSL surpass those obtained from the original Radon Transform"?

**Questions For Authors:**

Given my comments above, the proposed nonlinear tree SW variants seem to be incremental extensions of existing methods. This lack of novelty, in some sense, would not be problematic if the contributions yield clear advantages on other aspects, for instance here in terms of practical performance. However, in my opinion, this is not sufficiently achieved in the current work.

Therefore, I would appreciate if the authors could:
- Explicitly highlight the technical challenges involved in developing this approach or in establishing its theoretical guarantees, particularly in comparison to prior work.
- Design a simpler, more interpretable experiment (e.g.,  based on synthetic data) where the advantages of their proposed methods are more clearly demonstrated.

**Relation To Broader Scientific Literature:**

The key contribution of this paper is the combination of non-linear projections in sliced optimal transport with the tree structure. Both the theoretical results and experimental designs build on existing literature in this area, and the technical challenges involved in integrating these prior works are not sufficiently emphasized.

**Theoretical Claims:**

I skimmed through the proofs in the supplementary document. The theoretical claims appear to be adaptations of results from the literature on generalized/augmented SW and tree SW, with these extensions facilitated by the linear operations in the tree structure.

The authors focus on the first-order ($p = 1$) case, justifying this choice by stating: "For simplicity, the focus is on measures with a finite first moment, while measures with a finite $p$th-moment are treated analogously." (l.85-88) However, the treatment of $p>1$ is not as trivial as suggested when establishing the metric properties, injectivity, and well-definedness.

---

> ### Author Rebuttal · Authors · 2025-03-31
>
> We direct the Reviewer to Table R1-4 and Figure R1 at https://sites.google.com/view/nonlinear-tsw-2.
>
> **Q1. Explicitly highlight ... to prior work.**
>
> **Answer Q1.** The key technical challenge in developing this approach lies in proving the injectivity of the proposed Radon Transforms, which ensures that the resulting distances qualify as proper metrics. This differs from prior work on TSW, such as [2]. Compared to SW approaches—including those that also employ nonlinear projection techniques like [1]—the difference is even more pronounced due to the introduction of splitting maps $\alpha$.
>
> Given these factors, we believe our approach to addressing these challenges is non-trivial.
>
> **Q2. The notions of well-definedness ... Radon Transform variants".**
>
> **Answer Q2.** The two mentioned properties above hold under certain assumptions on the functions $g$ and $h$, such as continuity, smoothness, and injectivity. These conditions have been discussed in previous works (see lines 134–144).
>
> In our work, we do not elaborate on these general assumptions; instead, we provide specific types of functions that satisfy the necessary conditions for our method. A detailed discussion of why these functions meet the required properties are presented Section 4.2 and Appendix B.
>
> Injectivity is typically required for variants of the Radon Transform because it ensures that the derived metrics are proper metrics rather than pseudo-metrics.
>
> **Q3. In what sense ... Radon Transform?**
>
> **Q4. Why does $\alpha$ ... being referred to**
>
> **Answer Q3+Q4.** Previous works on TSW [2, 3, 4] show that replacing lines in the SW framework with tree structures via the splitting mechanism $\alpha$ consistently enhances performance, even with the same number of projections.
>
> Notably, $\alpha$ is independent of SW components, suggesting that other SW improvements could be integrated into the tree-sliced framework. However, verifying well-definedness and injectivity in this new setting requires novel analytical approaches.
>
> **Q5. Design a simpler ... clearly demonstrated.**
>
> **Answer Q5.**   The non-linearity in SpatialTSW complicates the design of settings where it outperforms other variants. Empirically, SpatialTSW matches Db-TSW (see Tables 1, 2, and R1), suggesting it as a drop-in replacement for Db-TSW with potential performance gains.
>
> We provide intuition for when CircularTSW and CircularTSW$_{r=0}$ may outperform other variants. Since these distances rely on the $L_2$ norm for projection, they are likely to excel when the $L_2$ norms of the data are diversely distributed. We validate this advantage over Db-TSW and SpatialTSW in Table R2 on a synthetic dataset. The improvement is more pronounced for high-dimensional data (large $d$), indicating that variants using circular defining functions perform well while others struggle in such settings.
>
> **Q6. Across all experiments, ... image generation quality.**
>
> **Answer Q6.** In our paper, uncertainty quantification for Table 2 is provided in the Appendix (see Table 7). We also include uncertainty quantification for the Diffusion Model and SWAE experiments in Tables R3 and R4, respectively.
>
> We provide a new qualitative result for Point Cloud Gradient Flow, visualizing the faster convergence of SpatialTSW in Figure R1.
>
> **Q7. The authors focus on the first-order ... and well-definedness.**
>
> **Answer Q7.** For $p>1$, the proposed approach can be extended. However, the Tree-Wasserstein distance with $p>1$ lacks a closed-form solution (see [5]). A meaningful alternative is provided by Sobolev Transport (ST) [6], which offers a closed-form solution and has been applied in the tree-sliced framework, as discussed in [3, Eq. (15)].
>
> Although TSW works such as [2] and [4] do not explicitly address this aspect, their implementations support arbitrary $p>1$. We omit the complex ST literature to reduce presentation complexity while keeping implementation flexible, and thus chose not to include it.
>
> Due to space constraints, we strongly encourage the Reviewer to refer to the ST literature [6]. Based on its properties, our extension to the $p>1$ case satisfies the theoretical guarantees previously discussed.
>
> **Q8. The results in Tables 1 and 2 ... is negligible**
>
> **Answer Q8.** We acknowledge that the reduction in computation time in Table 2 is marginal, as this experiment uses a toy synthetic dataset. However, in Table 1, when applied to real-world Diffusion Models, CircularTSW$_{r=0}$ reduces the total training time to 105.5 hours compared to 131 hours for Db-TSW over 1800 epochs, saving 25.5 hours of computation time.
>
> ---
> We thank the Reviewer for the constructive feedback, as well as for pointing out typos and missing references, which we will address. If our clarifications are satisfactory, we kindly ask you to consider raising the score. We are happy to address any further concerns in the next discussion stage.
>
> ---
> *References.* Kindly refer to **References** in our response to Reviewer jNth.

---

### Official Review · Reviewer_m8Ge · 2025-03-14

**Overall Recommendation:** 3

**Summary:**

The authors introduce several new variants of tree-sliced Wasserstein distance, which was introduced in [TPTLN '24]. This is done via two new proposed Radon transforms: (1) the generalized Radon transform on systems of lines and (2) the spatial Radon transform on systems of lines. Using their new Radon transforms, the authors define two variants tree-sliced Wasserstein distance: (1) circular tree-sliced Wasserstein (CircularTSW) distance and (2) spatial tree-sliced Wasserstein (SpatialTSW) distance. Unlike previous work, which used linear projections to construct tree-sliced Wasserstein distance, CircularTSW and SpatialTSW incorporate non-linear projections.

"Tree-sliced wasserstein distance on a system of lines" [TPTLN '24]

### Update after rebuttal

Thanks to the authors for their response. I maintain my score.

**Claims And Evidence:**

All claims/theorems are supported by proofs.

**Essential References Not Discussed:**

I do not know of any essential references which are not discussed.

**Experimental Designs Or Analyses:**

I checked the experimental design and it seems reasonable to me.

**Methods And Evaluation Criteria:**

The benchmark datasets seem reasonable to me.

**Other Comments Or Suggestions:**

You use the abbreviation for Circular Radon Transform on Systems of Lines (CRTSL) but you forgot to add the parenthetical defining the abbreviation above Eq. 10.

**Other Strengths And Weaknesses:**

Strengths: The authors present a novel extension of the previous tree-sliced Wasserstein distance to use non-linear projections. These new tree-sliced Wasserstein distance variants may help better encode topological information than tree-sliced Wasserstein with linear projection. They also present extensive experiments that highlight the strength of their new tree-sliced Wasserstein variants.

Weaknesses: Given that there are so many variants of TSW, the authors could maybe motivate a bit more why/when CircularTSW or SpatialTSW will outperform SW or TSW with linear projection.

**Questions For Authors:**

1. Is there any intuition for when each tree-sliced Wasserstein distance should be used? It seems now that is a large zoo of tree-sliced Wasserstein distances but I am unsure when I should use one tree-sliced Wasserstein distance over another.

2. How should I select $r$ for CircularTSW? How does the choice $r$ affect performance empirically?

3. It seems like this tree-sliced Wasserstein distance on systems of lines can be very general. Is there a way to define a general tree-sliced framework that will include both CRT and SRT?

**Relation To Broader Scientific Literature:**

This paper is a follow up to [TPTLN '24] and [TCNLN '25]. While [TPTLN '24] introduces the general tree-sliced Wasserstein distance on systems of lines, this work extends it by introducing generalized Radon transforms on systems of lines and defining the corresponding tree-sliced Wasserstein distances.  The current paper extends previous work by integrating nonlinear projection mechanisms, which allow for more flexible and expressive transformations of probability measures.

"Distance-based tree sliced Wasserstein distance" [TCNLN '25]

**Theoretical Claims:**

I did not check the proofs.

---

> ### Author Rebuttal · Authors · 2025-03-31
>
> We direct the Reviewer to Table R1-2 at https://sites.google.com/view/nonlinear-tsw.
>
> **W1. [...] (many variants of TSW) outperform SW or TSW with linear projection**
>
> **Q1. [...] (intuition) one (TSW) over another**
>
> **Answer.** Our motivation for proposing the non-linear projection framework is inspired by Generalized Sliced-Wasserstein (GSW) [1], which also includes both Circular and Spatial variants. It is underexplored in prior studies that among the three versions—original SW, SpatialSW, and CircularSW,  which variant is most suitable for a given task.
>
> This suggests that among the corresponding TSW variants—Db-TSW [2], CircularTSW, and Spatial TSW—there is no guarantee that the versions with non-linear projections will consistently outperform the linear-projection TSW. However, the two new distance variants each offer distinct advantages over standard TSW, as outlined below:
>
> - The definition of SpatialTSW subsumes Db-TSW as a special case when the function $h$ is the identity map (see lines 122–124). This implies that models leveraging SpatialTSW have, in theory, greater representational capacity than those using Db-TSW. A similar relationship holds between the corresponding SW variants.
> - The definition of CircularTSW is theoretically non-comparable to Db-TSW due to their fundamentally different constructions. However, $CircularTSW_{r=0}$ offers improved runtime efficiency. This benefit does not hold in the SW context, where $CircularSW_{r=0}$ performs poorly (see lines 252–261). One reason is that $CircularSW_{r=0}$ defines only a pseudo-metric, while $CircularTSW_{r=0}$ is a true metric.
>
> Our framework offers greater flexibility by enabling a broader selection of distance functions. However, in Machine Learning, predicting the best variant for a task often requires empirical experimentation. Table R1 shows that both Db-TSW and SpatialTSW perform well, but the non-linearity in SpatialTSW makes it hard to determine in advance which variant is better suited for a given task.
>
> We offer intuition for selecting CircularTSW and $CircularTSW_{r=0}$. Since these distances rely on the $L_2$ norm for the projection step, they are likely to perform well when the $L_2$ norms of the data are diversely distributed. We validate this advantage over Db-TSW and SpatialTSW in Table R2, where the distribution of $L_2$ norms is uniform. We speculate that this property explains why CircularTSW performs effectively for the Diffusion experiment (Table 1).
>
> To the best of our knowledge, Db-TSW [2] is the only tree-sliced distance effectively suited for large-scale generative tasks involving transport from a training measure to a target measure in Euclidean space. Previously, [3] presents a basic and limited version of [2], primarily emphasizing the constructive aspects of the tree-sliced approach, which serve as foundational groundwork. Meanwhile, [4] explores the method in a spherical setting. Other works on Tree-Sliced Wasserstein (TSW), such as [5], [7], and others, are mainly designed for classification tasks and are not applicable to generative settings. This limitation arises because these methods rely on a clustering-based framework for computing slices—which is theoretically unsuitable (as the clustering must be recomputed each time the training measure is updated, **rendering previous clustering results irrelevant**) and empirically inefficient (since clustering is significantly more computationally expensive than linear or non-linear projection methods).
>
> **Q2. [...] (select  $r$) affect performance empirically?**
>
> **Answer Q2.** Selecting the optimal hyperparameter, such as $r$ for CircularTSW, is challenging and often requires empirical tuning. Intuitively, $r$ should be large enough to ensure diverse projections onto the lines but should not exceed the data's magnitude. For normalized data, we suggest starting with $r = \frac{1}{\sqrt{d}}$ and tuning from there.
>
> **Q3. [...] (include) CRT and SRT?**
>
> **Answer Q3.** The TSW-SL [3] offers more general tree structure than the concurrent-lines tree structure used in [2]. However, this generality comes at the cost of runtime efficiency, as the concurrent-lines structure allows a GPU-friendly implementation. Since efficiency is a priority in our work, we adopt the concurrent-lines structure.
>
> When extending TSW-SL to non-linear projections, we initially believed SRT could apply to general tree structures, as key properties like injectivity are preserved. However, this does not seem to hold for CRT, since parts of the proof that CircularTSW defines a valid metric rely on the specific concurrent-lines structure.
>
> ---
> We thank the Reviewer for the constructive feedback and for pointing out the typos, which we will address. If our clarifications are satisfactory, we kindly ask you to consider raising the score. We are happy to address any further concerns in the next discussion stage.
>
> ---
> *References.* Kindly refer to **References** in our response to Reviewer jNth.

---

### Decision · Program_Chairs · 2025-05-01

**Decision:**

Accept (poster)

**Comment:**

This paper proposes a nonlinear extension of Tree-sliced Wasserstein distance on a system of lines. More specifically, the generalized Radon transform on systems of lines and the spatial Radon transform on systems of lines are proposed. Through experiments, it demonstrates that the proposed method works well on Gradient flow, SSL, and encoder tasks.

Note that, for the tree Wasserstein distance, the original work was introduced by Indyk et al. 2003. However, it is completely missing and this is not fair to the authors of the original work. For the final version, I strongly request adding detailed discussion with the previous work including Indyk et al. 2003.
[1] Piotr Indyk, Nitin Thaper. Fast image retrieval via embeddings, 2003.

Overall, this is a solid paper, and I vote for acceptance if the above concern is clearly addressed in the final version.